# REGULARIZED LEARNING FOR DOMAIN ADAPTATION UNDER LABEL SHIFTS

**Kamyar Azizzadenesheli**
University of California, Irvine
kazizzad@uci.edu

**Anqi Liu**
California Institute of Technology
anqiliu@caltech.edu

**Fanny Yang**
Institute of Theoretical Studies, ETH Zürich
fan.yang@stat.math.ethz.ch

**Animashree Anandkumar**
California Institute of Technology
anima@caltech.edu

## ABSTRACT

We propose Regularized Learning under Label shifts (RLLS), a principled and a practical domain-adaptation algorithm to correct for shifts in the label distribution between a source and a target domain. We first estimate importance weights using labeled source data and unlabeled target data, and then train a classifier on the weighted source samples. We derive a generalization bound for the classifier on the target domain which is independent of the (ambient) data dimensions, and instead only depends on the complexity of the function class. To the best of our knowledge, this is the first generalization bound for the label-shift problem where the labels in the target domain are not available. Based on this bound, we propose a regularized estimator for the small-sample regime which accounts for the uncertainty in the estimated weights. Experiments on the CIFAR-10 and MNIST datasets show that RLLS improves classification accuracy, especially in the low sample and large-shift regimes, compared to previous methods.

## 1 INTRODUCTION

When machine learning models are employed "in the wild", the distribution of the data of interest(*target* distribution) can be significantly shifted compared to the distribution of the data on which the model was trained (*source* distribution). In many cases, the publicly available large-scale datasets with which the models are trained do not represent and reflect the statistics of a particular dataset of interest. This is for example relevant in managed services on cloud providers used by clients in different domains and regions, or medical diagnostic tools trained on data collected in a small number of hospitals and deployed on previously unobserved populations and time frames.

There are various ways to approach distribution shifts between a source data distribution $\mathbb{P}$ and a target data distribution $\mathbb{Q}$. If we denote input variables as $x$ and output variables as $y$, we consider the two following settings: (i) Covariate shift, which

| Covariate Shift | Label Shift |
|---|---|
| $p(x) \neq q(x)$ | $p(y) \neq q(y)$ |
| $p(y\|x) = q(y\|x)$ | $p(x\|y) = q(x\|y)$ |

assumes that the conditional output distribution is invariant: $p(y|x) = q(y|x)$ between source and target distributions, but the source distribution $p(x)$ changes. (ii) Label shift, where the conditional input distribution is invariant: $p(x|y) = q(x|y)$ and $p(y)$ changes from source to target. In the following, we assume that both input and output variables are observed in the source distribution whereas only input variables are available from the target distribution.

While covariate shift has been the focus of the literature on distribution shifts to date, label-shift scenarios appear in a variety of practical machine learning problems and warrant a separate discussion as well. In one setting, suppliers of machine-learning models such as cloud providers have large resources of diverse data sets (source set) to train the models, while during deployment, they have no control over the proportion of label categories.

In another setting of e.g. medical diagnostics, the disease distribution changes over locations and time. Consider the task of diagnosing a disease in a country with bad infrastructure and little data,

based on reported symptoms. Can we use data from a different location with data abundance to diagnose the disease in the new target location in an efficient way? How many labeled source and unlabeled target data samples do we need to obtain good performance on the target data?

Apart from being relevant in practice, label shift is a computationally more tractable scenario than covariate shift which can be mitigated. The reason is that the outputs $y$ typically have a much lower dimension than the inputs $x$. Labels are usually either categorical variables with a finite number of categories or have simple well-defined structures. Despite being an intuitively natural scenario in many real-world application, even this simplified model has only been scarcely studied in the literature. Zhang et al. (2013) proposed a kernel mean matching method for label shift which is not computationally feasible for large-scale data. The approach in Lipton et al. (2018) is based on importance weights that are estimated using the confusion matrix (also used in the procedures of Saerens et al. (2002); McLachlan (2004)) and demonstrate promising performance on large-scale data. Using a black-box classifier which can be biased, uncalibrated and inaccurate, they first estimate importance weights $q(y)/p(y)$ for the source samples and train a classifier on the weighted data. In the following we refer to the procedure as *black box shift learning* (BBSL) which the authors proved to be effective for large enough sample sizes.

However, there are three relevant questions which remain unanswered by their work: How to estimate the importance weights in low sample setting, What are the generalization guarantees for the final predictor which uses the weighted samples? How do we deal with the uncertainty of the weight estimation when only few samples are available? This paper aims to fill the gap in terms of both theoretical understanding and practical methods for the label shift setting and thereby move a step closer towards having a more complete understanding on the general topic of supervised learning for distributionally shifted data. In particular, our goal is to find an efficient method which is applicable to large-scale data and to establish generalization guarantees.

Our contribution in this work is trifold. Firstly, we propose an efficient weight estimator for which we can obtain good statistical guarantees without a requirement on the problem-dependent minimum sample complexity as necessary for BBSL. In the BBSL case, the estimation error can become arbitrarily large for small sample sizes. Secondly, we propose a novel regularization method to compensate for the high estimation error of the importance weights in low target sample settings. It explicitly controls the influence of our weight estimates when the target sample size is low (in the following referred to as the low sample regime). Finally, we derive a dimension-independent generalization bound for the final Regularized Learning under Label Shift (RLLS) classifier based on our weight estimator. In particular, our method improves the weight estimation error and excess risk of the classifier on reweighted samples by a factor of $k \log(k)$, where $k$ is the number of classes, i.e. the cardinality of $\mathcal{Y}$.

In order to demonstrate the benefit of the proposed method for practical situations, we empirically study the performance of RLLS and show weight estimation as well as prediction accuracy comparison for a variety of shifts, sample sizes and regularization parameters on the CIFAR-10 and MNIST datasets. For large target sample sizes and large shifts, when applying the regularized weights fully, we achieve an order of magnitude smaller weight estimation error than baseline methods and enjoy at most 20% higher accuracy and F-1 score in corresponding predictive tasks. For low target sample sizes, applying regularized weights partially also yields an accuracy improvement of at least 10% over fully weighted and unweighted methods.

## 2   REGULARIZED LEARNING OF LABEL SHIFTS (RLLS)

Formally let us the short hand for the marginal probability mass functions of $Y$ on finite $\mathcal{Y}$ with respect to $\mathbb{P}, \mathbb{Q}$ as $p, q : [k] \to [0, 1]$ with $p(i) = \mathbb{P}(Y = i)$, and $q(i) = \mathbb{Q}(Y = i)$ for all $i \in [k]$, representable by vectors in $\mathbb{R}_+^k$ which sum to one. In the label shift setting, we define the importance weight vector $w \in \mathbb{R}^k$ between these two domains as $w(i) = \frac{q(i)}{p(i)}$. We quantify the shift using the exponent of the infinite and second order Renyi divergence as follows

$$d_\infty(q||p) := \sup_i \frac{q(i)}{p(i)} \quad , \quad and \quad d(q||p) := \mathbb{E}_{Y \sim \mathbb{Q}} \left[ w(Y)^2 \right] = \sum_i^k q(i) \frac{q(i)}{p(i)}.$$

Given a hypothesis class $\mathcal{H}$ and a loss function $\ell : \mathcal{Y} \times \mathcal{Y} \to [0, 1]$, our aim is to find the hypothesis $h \in \mathcal{H}$ which minimizes

$$\mathcal{L}(h) = \mathbb{E}_{X,Y \sim \mathbb{Q}}\left[\ell(Y, h(X))\right] = \mathbb{E}_{X,Y \sim \mathbb{P}}\left[w(Y)\ell(Y, h(X))\right]$$

In the usual finite sample setting however, $\mathcal{L}$ unknown and we observe samples $\{(x_j, y_j)\}_{j=1}^n$ from $\mathbb{P}$ instead. If we are given the vector of importance weights $w$ we could then minimize the empirical loss with importance weighted samples defined as

$$\mathcal{L}_n(h) = \frac{1}{n}\sum_{j=1}^n w(y_j)\ell(y_j, h(x_j))$$

where $n$ is the number of available observations drawn from $\mathbb{P}$ used to learn the classifier $h$. As $w$ is unknown in practice, we have to find the minimizer of the empirical loss with *estimated* importance weights

$$\mathcal{L}_n(h; \widehat{w}) = \frac{1}{n}\sum_{j=1}^n \widehat{w}(y_j)\ell(y_j, h(x_j)) \tag{1}$$

where $\widehat{w}$ are estimates of $w$. Given a set $D_p$ of $n_p$ samples from the source distribution $\mathbb{P}$, we first divide it into two sets where we use $(1-\beta)n_p$ samples in set $D_p^{\text{weight}}$ to compute the estimate $\widehat{w}$ and the remaining $n = \beta n_p$ in the set $D_p^{\text{class}}$ to find the classifier which minimizes the loss (1), i.e. $\widehat{h}_{\widehat{w}} = \arg\min_{h \in \mathcal{H}} \mathcal{L}_n(h; \widehat{w})$. In the following, we describe how to estimate the weights $\widehat{w}$ and provide guarantees for the resulting estimator $\widehat{h}_{\widehat{w}}$.

**Plug-in weight estimation** The following simple correlation between the label distributions $p, q$ was noted in Lipton et al. (2018): for a fixed hypothesis $h$, if for all $y \in \mathcal{Y}$ it holds that $q(y) \geq 0 \implies p(y) \geq 0$, we have

$$q_h(i) := \mathbb{Q}(h(X) = i) = \sum_{j=1}^k \mathbb{Q}(h(X) = i | Y = j)q(j) = \sum_{j=1}^k \mathbb{P}(h(X) = i | Y = j)q(j)$$

$$= \sum_{j=1}^k \mathbb{P}(h(X) = i, Y = j)\frac{q(j)}{p(j)} = \sum_{j=1}^k \mathbb{P}(h(X) = i, Y = j)w_j$$

for all $i, j \in \mathcal{Y}$. This can equivalently be written in matrix vector notation as

$$q_h = C_h w, \tag{2}$$

where $C_h$ is the confusion matrix with $[C_h]_{i,j} = \mathbb{P}(h(X) = i, Y = j)$ and $q_h$ is the vector which represents the probability mass function of $h(X)$ under distribution $\mathbb{Q}$. The requirement $q(y) \geq 0 \implies p(y) \geq 0$ is a reasonable condition since without any prior knowledge, there is no way to properly reason about a class in the target domain that is not represented in the source domain.

In reality, both $q_h$ and $C_h$ can only be estimated by the corresponding finite sample averages $\widehat{q}_h, \widehat{C}_h$. Lipton et al. (2018) simply compute the inverse of the estimated confusion matrix $\widehat{C}_h$ in order to estimate the importance weight, i.e. $\widehat{w} = \widehat{C}_h^{-1}\widehat{q}_h$. While $C_h^{-1}\widehat{q}_h$ is a statistically efficient estimator, $\widehat{w}$ with estimated $\widehat{C}_h^{-1}$ can be arbitrarily bad since $\widehat{C}_h^{-1}$ can be arbitrary close to a singular matrix especially for small sample sizes and small minimum singular value of the confusion matrix. Intuitively, when there are very few samples, the weight estimation will have high variance in which case it might be better to avoid importance weighting altogether. Furthermore, even when the sample complexity in Lipton et al. (2018), unknown in practice, is met, the resulting error of this estimator is linear in $k$ which is problematic for large $k$.

We therefore aim to address these shortcomings by proposing the following two-step procedure to compute importance weights. In the case of no shift we have $w = \mathbf{1}$ so that we define the amount of weight shift as $\theta = w - \mathbf{1}$. Given a "decent" black box estimator which we denote by $h_0$, we make the final classifier less sensitive to the estimation performance of $C$ (i.e. regularize the weight estimate) by

1. calculating the measurement error adjusted $\widehat{\theta}$ (described in Section 2.1 for $h_0$) and

2. computing the regularized weight $\widehat{w} = \mathbf{1} + \lambda\widehat{\theta}$ where $\lambda$ depends on the sample size $(1-\beta)n_p$.

By "decent" we refer to a classifier $h_0$ which yields a full rank confusion matrix $C_{h_0}$. A trivial example for a non-"decent" classifier $h_0$ is one that always outputs a fixed class. As it does not capture any characteristics of the data, there is no hope to gain any statistical information without any prior information.

## 2.1 Estimator correcting for finite sample errors

Both the confusion matrix $C_{h_0}$ and the label distribution $q_{h_0}$ on the target for the black box hypothesis $h_0$ are unknown and we are instead only given access to finite sample estimates $\widehat{C}_{h_0}, \widehat{q}_{h_0}$. In what follows all empirical and population confusion matrices, as well as label distributions, are defined with respect to the hypothesis $h = h_0$. For notation simplicity, we thus drop the subscript $h_0$ in what follows. The reparameterized linear model (2) with respect to $\theta$ then reads

$$b := q - C\mathbf{1} = C\theta$$

with corresponding finite sample quantity $\widehat{b} = \widehat{q} - \widehat{C}\mathbf{1}$. When $\widehat{C}$ is near singular, the estimation of $\theta$ becomes unstable. Furthermore, large values in the true shift $\theta$ result in large variances. We address this problem by adding a regularizing $\ell_2$ penalty term to the usual loss and thus push the amount of shift towards 0, a method that has been proposed in (Pires & Szepesvári, 2012). In particular, we compute

$$\widehat{\theta} = \arg\min_{\theta} \|\widehat{C}\theta - \widehat{b}\|_2 + \Delta_C\|\theta\|_2 \tag{3}$$

Here, $\Delta_C$ is a parameter which will eventually be high probability upper bounds for $\|\widehat{C} - C\|_2$. Let $\Delta_b$ also denote the high probability upper bounds for $\|\widehat{b} - b\|_2$.

**Lemma 1** *For $\widehat{\theta}$ as defined in equation* (3), *we have with probability at least* $1 - \delta$ *that*[1]

$$\|\widehat{\theta} - \theta\|_2 \leq \epsilon_\theta(n_p, n_q, \|\theta\|_2, \delta)$$

*where*

$$\epsilon_\theta(n_p, n_q, \|\theta\|_2, \delta) := \mathcal{O}\left(\frac{1}{\sigma_{min}}\left(\|\theta\|_2\sqrt{\frac{\log(k/\delta)}{(1-\beta)n_p}} + \sqrt{\frac{\log(1/\delta)}{(1-\beta)n_p}} + \sqrt{\frac{\log(1/\delta)}{n_q}}\right)\right).$$

The proof of this lemma can be found in Appendix B.1. A couple of remarks are in order at this point. First of all, notice that the weight estimation procedure (3) does not require a minimum sample complexity which is in the order of $\sigma_{\min}^{-2}$ to obtain the guarantees for BBSL. This is due to the fact that errors in the covariates are accounted for. In order to directly see the improvements in the upper bound of Lemma 1 compared to Theorem 3 in Lipton et al. (2018), first observe that in order to obtain their upper bound with a probability of at least $1 - \delta$, it is necessary that $3kn_p^{-10} + 2kn_q^{-10} \leq \delta$. As a consequence, the upper bound in Theorem 3 of Lipton et al. (2018) is bigger than $\frac{1}{3\sigma_{\min}}\left(\|\theta\|_2\sqrt{\frac{\log(3k/\delta)}{n_p}} + \sqrt{\frac{k\log(2k/\delta)}{n_q}}\right)$. Thus Lemma 1 improves upon the previous upper bound by a factor of $k$.

Furthermore, as in Lipton et al. (2018), this result holds for *any* black box estimator $h_0$ which enters the bound via $\sigma_{\min}(C_{h_0})$. We can directly see how a good choice of $h_0$ helps to decrease the upper bound in Lemma 1. In particular, if $h_0$ is an ideal estimator, and the source set is balanced, $C$ is the unit matrix with $\sigma_{\min} = 1/k$. In contrast, when the model $h_0$ is uncertain, the singular value $\sigma_{\min}$ is close to zero.

Moreover, for least square problems with Gaussian measurement errors in both input and target variables, it is standard to use regularized total least squares approaches which requires a singular value decomposition. Finally, our choice for the alternative estimator in Eq. 3 with norm instead of norm squared regularization is motivated by the cases with large shifts $\theta$, where using the squared norm may shrink the estimate $\widehat{\theta}$ too much and away from the true $\theta$.

---

[1]Throughout the paper, $\mathcal{O}$ hides universal constant factors. Furthermore, we use $\mathcal{O}\left(\cdot + \cdot\right)$ for short to denote $\mathcal{O}\left(\cdot\right) + \mathcal{O}\left(\cdot\right)$.

---

**Algorithm 1** Regularized Learning of Label Shift (RLLS)

---

1: Input: source set $D_p$, $D_q$, $\theta_{\max}$, estimate of $\sigma_{\min}$, black box estimator $h_0$, model class $\mathcal{H}$
2: Determine optimal split ratio $\beta^\star$ and regularizer $\lambda^\star$ by minimizing the RHS of Eq. (6) using an estimate of $\sigma_{\min}$
3: Randomly partition source set $D_p$ into $D_p^{\text{class}}$, $D_p^{\text{weight}}$ such that $|D_p^{\text{class}}| = \beta^\star n_p =: n$
4: Compute $\widehat{\theta}$ using Eq. (3) and $\widehat{w} := 1 + \lambda^\star \widehat{\theta}$
5: Minimize the importance weighted empirical loss to obtain the weighted estimator

$$\widehat{h}_{\widehat{w}} = \arg\min_{h \in \mathcal{H}} \mathcal{L}_n(h; \widehat{w}), \qquad where \qquad \mathcal{L}_n(h; \widehat{w}) = \frac{1}{n} \sum_{(x,y) \in D_p^{\text{class}}} \widehat{w}(y)\ell(y, h(x))$$

6: Deploy $\widehat{h}_{\widehat{w}}$ if the risk is acceptable

---

## 2.2 REGULARIZED ESTIMATOR AND GENERALIZATION BOUND

When a few samples from the target set are available or the label shift is mild, the estimated weights might be too uncertain to be applied. We therefore propose a regularized estimator defined as follows

$$\widehat{w} = \mathbf{1} + \lambda\widehat{\theta}. \tag{4}$$

Note that $\widehat{w}$ implicitly depends on $\lambda$, and $\beta$. By rewriting $\widehat{w} = (1 - \lambda)\mathbf{1} + \lambda(\mathbf{1} + \widehat{\theta})$, we see that intuitively $\lambda$ closer to 1 the more reason there is to believe that $\mathbf{1} + \widehat{\theta}$ is in fact the true weight.

Define the set $\mathcal{G}(\ell, \mathcal{H}) = \{g_h(x, y) = w(y)\ell(h(x), y) : h \in \mathcal{H}\}$ and its Rademacher complexity measure

$$\mathcal{R}_n(\mathcal{G}) := \mathbb{E}_{(X_i, Y_i) \sim \mathbb{P}: i \in [n]} \left[ \mathbb{E}_{\xi_i : i \in [n]} \frac{1}{n} \left[ \sup_{h \in \mathcal{H}} \sum_{i=1}^{n} \xi_i g_h(X_i, h(Y_i)) \right] \right]$$

with $\xi_i$, $\forall i$ as the Rademacher random variables (see e.g. Bartlett & Mendelson (2002)). We can now state a generalization bound for the classifier $\widehat{h}_{\widehat{w}}$ in a general hypothesis class $\mathcal{H}$, which is trained on source data with the estimated weights defined in equation (4).

**Theorem 1 (Generalization bound for $\widehat{h}_{\widehat{w}}$)** *Given $n_p$ samples from the source data set and $n_q$ samples from the target set, a hypothesis class $\mathcal{H}$ and loss function $\ell$, the following generalization bound holds with probability at least $1 - 2\delta$*

$$\mathcal{L}(\widehat{h}_{\widehat{w}}) - \mathcal{L}(h^*) \leq \epsilon_{\mathcal{G}}(n_p, \delta, \beta) + (1 - \lambda)\|\theta\|_2 + \lambda\epsilon_\theta(n_p, n_q, \|\theta\|_2, \delta, \beta). \tag{5}$$

*where*

$$\epsilon_{\mathcal{G}}(n_p, \delta) := 2\mathcal{R}_n(\mathcal{G}) + \min\left\{ d_\infty(q\|p)\sqrt{\frac{\log(2/\delta)}{\beta n_p}}, \frac{2d_\infty(q\|p)\log(2/\delta)}{n} + \sqrt{2\frac{d(q\|p)\log(2/\delta)}{n}} \right\}.$$

The proof can be found in Appendix B.4. Additionally, we derive the analysis also for finite hypothesis classes in Appendix B.6 to provide more insight into the proof of general hypothesis classes. The size of $\mathcal{R}_n(\mathcal{G})$ is determined by the structure of the function class $\mathcal{H}$ and the loss $\ell$. For example for the $0/1$ loss, the VC dimension of $\mathcal{H}$ can be deployed to upper bound the Rademacher complexity.

The bound (5) in Theorem 1 holds for all choices of $\lambda$. In order to exploit the possibility of choosing $\lambda$ and $\beta$ to have an improved accuracy depending on the sample sizes, we first let the user define a set of shifts $\theta$ against which we want to be robust against, i.e. all shifts with $\|\theta\|_2 \leq \theta_{\max}$. For these shifts, we obtain the following upper bound

$$\mathcal{L}(\widehat{h}_{\widehat{w}}) - \mathcal{L}(h^*) \leq \epsilon_{\mathcal{G}}(n_p, \delta) + (1 - \lambda)\theta_{\max} + \lambda\epsilon_\theta(n_p, n_q, \theta_{\max}, \delta) \tag{6}$$

The bound in equation (6) suggests using Algorithm 1 as our ultimate label shift correction procedure. where for step 2 of the algorithm, we choose $\lambda^\star = 1$ whenever $n_q \geq \frac{1}{\theta_{\max}^2(\sigma_{\min} - \frac{1}{\sqrt{n_p}})^2}$ (hereby neglecting the log factors and thus dependencies on $k$) and 0 else. When using this rule, we

obtain $\mathcal{L}(\widehat{h}_{\widehat{w}}) - \mathcal{L}(h^*) \leq \epsilon_{\mathcal{G}}(n_p, \delta) + \min\{\theta_{\max}, \epsilon_\theta(n_p, n_q, \theta_{\max}, \delta)\}$ which is smaller than the unregularized bound for small $n_q, n_p$. Notice that in practice, we do not know $\sigma_{\min}$ in advance so that in Algorithm 1 we need to use an estimate of $\sigma_{\min}$, which could e.g. be the minimum eigenvalue of the empirical confusion matrix $\widehat{C}$ with an additional computational complexity of at most $O(k^3)$.

Figure 1 shows how the oracle thresholds vary with $n_q$ and $\sigma_{\min}$ when $n_p$ is kept fix. When the parameters are above the curves for fixed $n_p$, $\lambda$ should be chosen as 1 otherwise the samples should be unweighted, i.e. $\lambda = 0$. This figure illustrates that when the confusion matrix has small singular values, the estimated weights should only be trusted for rather high $n_q$ and high believed shifts $\theta_{\max}$. Although the overall statistical rate of the excess risk of the classifier does not change as a function of the sample sizes, $\theta_{\max}$ could be significantly smaller than $\epsilon_\theta$ when $\sigma_{\min}$ is very small and thus the accuracy in this regime could improve. Indeed we observe this to be the case empirically in Section 3.3.

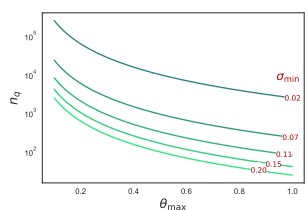

Figure 1: Given a $\sigma_{\min}$ and $\theta_{\max}$, $\lambda$ switches from 0 to 1 at a particular $n_q$. $n_p$ and $k$ are fixed.

In the case of slight deviation from the label shift setting, we expect the Alg. 1 to perform reasonably. For $d_e(q||p) := \mathbb{E}_{(X,Y)\sim\mathbb{Q}}\left[\left|1 - \frac{p(X|Y)}{q(X|Y)}\right|\right]$ as the deviation form label shift constraint, i.e., zero under label shift assumption, we have

**Theorem 2 (Drift in Label shift assumption)** *In the presence of $d_e(q||p)$ deviation from label shift assumption, the true importance weights $\omega(x,y) := \frac{q(x,y)}{p(x,y)}$, the RLLS generalizes as;*

$$\mathcal{L}(\widehat{h}_{\widehat{w}}, \omega) - \mathcal{L}(h^*; \omega) \leq \epsilon_{\mathcal{G}}(n_p, \delta) + (1-\lambda)\|\theta\|_2 + \lambda\epsilon_\theta(n_p, n_q, \|\theta\|_2, \delta) + 4(1-\lambda)d_e(q||p)$$

*with high probability. Proof in Appendix B.7.*

## 3 EXPERIMENTS

In this section we illustrate the theoretical analysis by running RLLS on a variety of artificially generated shifts on the MNIST (LeCun & Cortes, 2010) and CIFAR10 (Krizhevsky & Hinton, 2009) datasets. We first randomly separate the entire dataset into two sets (source and target pool) of the same size. Then we sample, unless specified otherwise, the same number of data points from each pool to form the source and target set respectively. We chose to have equal sample sizes to allow for fair comparisons across shifts.

There are various kinds of shifts which we consider in our experiments. In general we assume one of the source or target datasets to have uniform distribution over the labels. Within the non-uniform set, we consider three types of sampling strategies in the main text: the *Tweak-One shift* refers to the case where we set a class to have probability $p > 0.1$, while the distribution over the rest of the classes is uniform. The *Minority-Class Shift* is a more general version of *Tweak-One shift*, where a fixed number of classes $m$ to have probability $p < 0.1$, while the distribution over the rest of the classes is uniform. For the *Dirichlet shift*, we draw a probability vector $p$ from the Dirichlet distribution with concentration parameter set to $\alpha$ for all classes, before including sample points which correspond to the multinomial label variable according to $p$. Results for the tweak-one shift strategy as in Lipton et al. (2018) can be found in Section A.0.1.

After artificially shifting the label distribution in one of the source and target sets, we then follow algorithm 1, where we choose the black box predictor $h_0$ to be a two-layer fully connected neural network trained on (shifted) source dataset. Note that any black box predictor could be employed here, though the higher the accuracy, the more likely weight estimation will be precise. Therefore, we use different shifted source data to get (corrupted) black box predictor across experiments. If not noted, $h_0$ is trained using uniform data.

In order to compute $\widehat{\omega} = \mathbf{1} + \widehat{\theta}$ in Eq. (3), we call a built-in solver to directly solve the low dimensional problem $\min_\theta \|\widehat{C}\theta - \widehat{b}\|_2 + \Delta_C\|\theta\|_2$ where we empirically observer that 0.01 times of the true $\Delta_C$ yields in a better estimator on various levels of label shift pre-computed beforehand. It is worth noting that 0.001 makes the theoretical bound in Lemma. 1 $\mathcal{O}(1/0.01)$ times bigger. We thus treat it as a hyperparameter that can be chosen using standard cross validation methods. Finally, we train

a classifier on the source samples weighted by $\widehat{\omega}$, where we use a two-layer fully connected neural network for MNIST and a ResNet-18 (He et al., 2016) for CIFAR10.

We sample 20 datasets with the label distributions for each shift parameter. to evaluate the empirical mean square estimation error (MSE) and variance of the estimated weights $\mathbb{E}\|\widehat{w} - w\|_2^2$ and the predictive accuracy on the target set. We use these measures to compare our procedure with the black box shift learning method (BBSL) in Lipton et al. (2018). Notice that although KMM methods (Zhang et al., 2013) would be another standard baseline to compare with, it is not scalable to large sample size regimes for $n_p, n_q$ above $n = 8000$ as mentioned by Lipton et al. (2018).

## 3.1 WEIGHT ESTIMATION AND PREDICTIVE PERFORMANCE FOR SOURCE SHIFT

In this set of experiments on the CIFAR10 dataset, we illustrate our weight estimation and prediction performance for Tweak-One source shifts and compare it with BBSL. For this set of experiments, we set the number of data points in both source and target set to 10000 and sample from the two pools without replacement.

Figure 2 illustrates the weight estimation alongside final classification performance for Minority-Class source shift of CIFAR10. We created shifts with $\rho > 0.5$. We use a fixed black-box classifier that is trained on biased source data, with tweak-one $\rho = 0.5$. Observe that the MSE in weight estimation is relatively large and RLLS outperforms BBSL as the number of minority classes increases. As the shift increases the performance for all methods deteriorates. Furthermore, Figure 2 (b) illustrates how the advantage of RLLS over the unweighted classifier increases as the shift increases. Across all shifts, the RLLS based classifier yields higher accuracy than the one based on BBSL. Results for MNIST can be found in Section A.1.

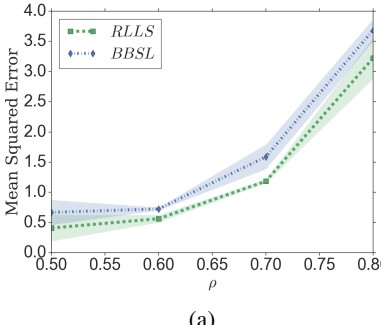 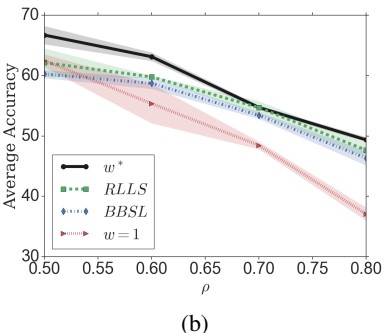

(a)                                          (b)

Figure 2: (a) Mean squared error in estimated weights and (b) accuracy on CIFAR10 for tweak-one shifted source and uniform target with $h_0$ trained using tweak-one shifted source data.

## 3.2 WEIGHT ESTIMATION AND PREDICTIVE PERFORMANCE FOR TARGET SHIFT

In this section, we compare the predictive performances between a classifier trained on unweighted source data and the classifiers trained on weighted loss obtained by the RLLS and BBSL procedure on CIFAR10. The target set is shifted using the Dirichlet shift with parameters $\alpha = [0.01, 0.1, 1, 10]$. The number of data points in both source and target set is 10000.

In the case of target shifts, larger shifts actually make the predictive task easier, such that even a constant majority class vote would give high accuracy. However it would have zero accuracy on all but one class. Therefore, in order to allow for a more comprehensive performance between the methods, we also compute the macro-averaged *F-1 score* by averaging the per-class quantity $2(\text{precision} \cdot \text{recall})/(\text{precision} + \text{recall})$ over all classes. For a class $i$, *precision* is the percentage of correct predictions among all samples predicted to have label $i$, while *recall* is the proportion of correctly predicted labels over the number of samples with true label $i$. This measure gives higher weight to the accuracies of minority classes which have no effect on the total accuracy.

Figure 3 depicts the MSE of the weight estimation (a), the corresponding performance comparison on accuracy (b) and F-1 score (c). Recall that the accuracy performance for low shifts is not comparable with standard CIFAR10 benchmark results because of the overall lower sample size chosen for the comparability between shifts. We can see that in the large target shift case for $\alpha = 0.01$, the F-1

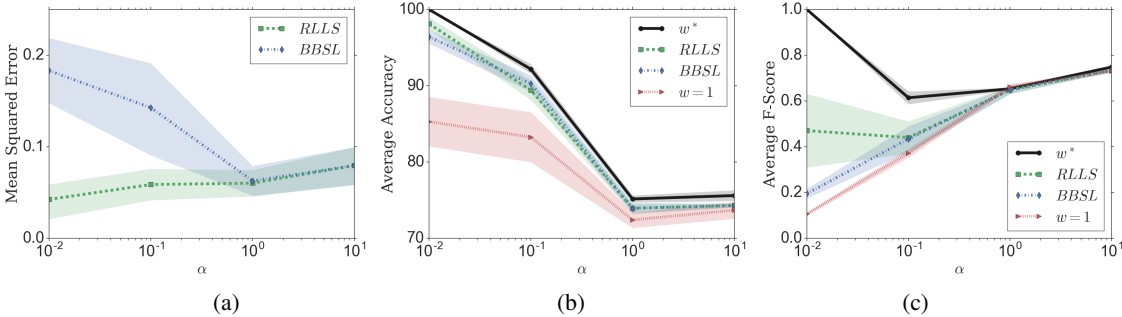

Figure 3: (a) Mean squared error in estimated weights, (b) accuracy and (c) F-1 score on CIFAR10 for uniform source and Dirichlet shifted target. Smaller $\alpha$ corresponds to bigger shift.

score for BBSL and the unweighted classifier is rather low compared to RLLS while the accuracy is high. As mentioned before, the reason for this observation and why in Figure 3 (b) the accuracy is higher when the shift is larger, is that the predictive task actually becomes easier with higher shift.

### 3.3 REGULARIZED WEIGHTS IN THE LOW SAMPLE REGIME FOR SOURCE SHIFT

In the following, we present the average accuracy of RLLS in Figure 4 as a function of the number of target samples $n_q$ for different values of $\lambda$ for small $n_q$. Here we fix the sample size in the source set to $n_p = 1000$ and investigate a Minority-Class source shift with fixed $p = 0.01$ and five minority classes.

A motivation to use intermediate $\lambda$ is discussed in Section 2.2, as $\lambda$ in equation (4) may be chosen according to $\theta_{\max}, \sigma_{\min}$. In practice, since $\theta_{\max}$ is just an upper bound on the true amount of shift $\|\theta\|_2$, in some cases $\lambda$ should in fact ideally be 0 when $\frac{1}{\theta_{\max}^2(\sigma_{\min}-\frac{1}{\sqrt{n_q}})^2} \leq n_q \leq \frac{1}{\|\theta\|_2(\sigma_{\min}-\frac{1}{\sqrt{n_q}})^2}$.

Thus for target sample sizes $n_q$ that are a little bit above the threshold (depending on the certainty of the belief how close to $\theta_{\max}$ the norm of the shift is believed to be), it could be sensible to use an intermediate value $\lambda \in (0, 1)$.

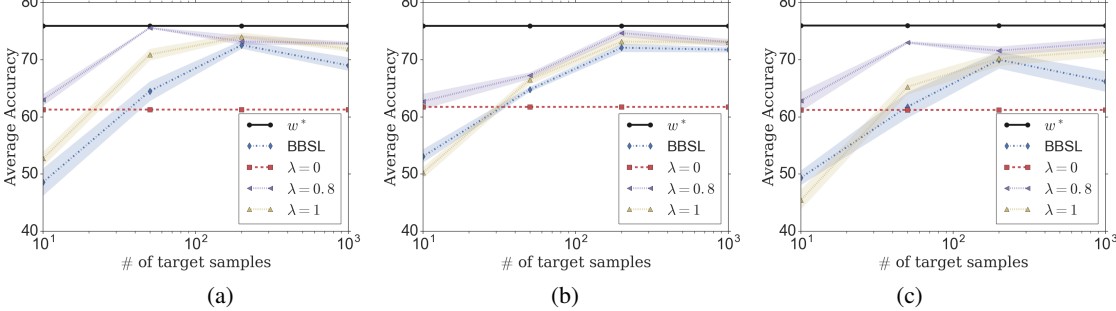

Figure 4: Performance on MNIST for Minority-Class shifted source and uniform target with various target sample size and $\lambda$ using (a) better predictor $h_0$ trained on tweak-one shifted source with $\rho = 0.2$, (b) neutral predictor $h_0$ with $\rho = 0.5$ and (c) corrupted predictor $h_0$ with $\rho = 0.8$.

Figure 4 suggests that unweighted samples (red) yield the best classifier for very few samples $n_q$, while for $10 \leq n_q \leq 500$ an intermediate $\lambda \in (0, 1)$ (purple) has the highest accuracy and for $n_q > 1000$, the weight estimation is certain enough for the fully weighted classifier (yellow) to have the best performance (see also the corresponding data points in Figure 2). The unweighted BBSL classifier is also shown for completeness. We can conclude that regularizing the influence of the estimated weights allows us to adjust to the uncertainty on importance weights and generalize well for a wide range of target sample sizes.

Furthermore, the different plots in Figure 4 correspond to black-box predictors $h_0$ for weight estimation which are trained on more or less corrupted data, i.e. have a better or worse conditioned

confusion matrix. The fully weighted methods with $\lambda = 1$ achieve the best performance faster with a better trained black-box classifier (a), while it takes longer for it to improve with a corrupted one (c). Furthermore, this reflects the relation between eigenvalue of confusion matrix $\sigma_{\min}$ and target sample size $n_q$ in Theorem 1. In other words, we need more samples from the target data to compensate a bad predictor in weight estimation. So the generalization error decreases faster with an increasing number of samples for good predictors.

In summary, our RLLS method outperforms BBSL in all settings for the common image datasets MNIST and CIFAR10 to varying degrees. In general, significant improvements compared to BBSL can be observed for large shifts and the low sample regime. A note of caution is in order: comparison between the two methods alone might not always be meaningful. In particular, there are cases when the estimator trained on unweighted samples outperforms both RLLS and BBSL. Our extensive experiments for many different shifts, black box classifiers and sample sizes do not allow for a final conclusive statement about how weighting samples using our estimator affects predictive results for real-world data in general, as it usually does not fulfill the label-shift assumptions.

## 4 RELATED WORK

The covariate and label shift assumptions follow naturally when viewing the data generating process as a causal or anti-causal model (Schölkopf et al., 2012): With label shift, the label $Y$ causes the input $X$ (that is, $X$ is not a causal parent of $Y$, hence "anti-causal") and the causal mechanism that generates $X$ from $Y$ is independent of the distribution of $Y$. A long line of work has addressed the reverse causal setting where $X$ causes $Y$ and the conditional distribution of $Y$ given $X$ is assumed to be constant. This assumption is sensible when there is reason to believe that there is a true optimal mapping from $X$ to $Y$ which does not change if the distribution of $X$ changes. Mathematically this scenario corresponds to the covariate shift assumption.

Among the various methods to correct for covariate shift, the majority uses the concept of importance weights $q(x)/p(x)$ (Zadrozny, 2004; Cortes et al., 2010; Cortes & Mohri, 2014; Shimodaira, 2000), which are unknown but can be estimated for example via kernel embeddings (Huang et al., 2007; Gretton et al., 2009; 2012; Zhang et al., 2013; Zaremba et al., 2013) or by learning a binary discriminative classifier between source and target (Lopez-Paz & Oquab, 2016; Liu et al., 2017). A minimax approach that aims to be robust to the worst-case shared conditional label distribution between source and target has also been investigated (Liu & Ziebart, 2014; Chen et al., 2016). Sanderson & Scott (2014); Ramaswamy et al. (2016) formulate the label shift problem as a mixture of the class conditional covariate distributions with unknown mixture weights. Under the pairwise mutual irreducibility (Scott et al., 2013) assumption on the class conditional covariate distributions, they deploy the Neyman-Pearson criterion (Blanchard et al., 2010) to estimate the class distribution $q(y)$ which also investigated in the maximum mean discrepancy framework (Iyer et al., 2014).

Common issues shared by these methods is that they either result in a massive computational burden for large sample size problems or cannot be deployed for neural networks. Furthermore, importance weighting methods such as (Shimodaira, 2000) estimate the density (ratio) beforehand, which is a difficult task on its own when the data is high-dimensional. The resulting generalization bounds based on importance weighting methods require the second order moments of the density ratio $(q(x)/p(x))^2$ to be bounded, which means the bounds are extremely loose in most cases (Cortes et al., 2010).

Despite the wide applicability of label shift, approaches with global guarantees in high dimensional data regimes remain under-explored. The correction of label shift mainly requires to estimate the importance weights $q(y)/p(y)$ over the labels which typically live in a very low-dimensional space. Bayesian and probabilistic approaches are studied when a prior over the marginal label distribution is assumed (Storkey, 2009; Chan & Ng, 2005). These methods often need to explicitly compute the posterior distribution of $y$ and suffer from the curse of dimensionality. Recent advances as in Lipton et al. (2018) have proposed solutions applicable large scale data. This approach is related to Buck et al. (1966); Forman (2008); Saerens et al. (2002) in the low dimensional setting but lacks guarantees for the excess risk.

Existing generalization bounds have historically been mainly developed for the case when $\mathbb{P} = \mathbb{Q}$ (see e.g. Vapnik (1999); Bartlett & Mendelson (2002); Kakade et al. (2009); Wainwright (2019)).

Ben-David et al. (2010) provides theoretical analysis and generalization guarantees for distribution shifts when the H-divergence between joint distributions is considered, whereas Crammer et al. (2008) proves generalization bounds for learning from multiple sources. For the covariate shift setting, Cortes et al. (2010) provides a generalization bound when $q(x)/p(x)$ is known which however does not apply in practice. To the best of our knowledge our work is the first to give generalization bounds for the label shift scenario.

## 5 DISCUSSION

In this work, we establish the first generalization guarantee for the label shift setting and propose an importance weighting procedure for which no prior knowledge of $q(y)/p(y)$ is required. Although RLLS is inspired by BBSL, it leads to a more robust importance weight estimator as well as generalization guarantees in particular for the small sample regime, which BBSL does not allow for. RLLS is also equipped with a sample-size-dependent regularization technique and further improves the classifier in both regimes.

We consider this work a necessary step in the direction of solving shifts of this type, although the label shift assumption itself might be too simplified in the real world. In future work, we plan to also study the setting when it is slightly violated. For instance, $x$ in practice cannot be solely explained by the wanted label $y$, but may also depend on attributes $z$ which might not be observable. In the disease prediction task for example, the symptoms might not only depend on the disease but also on the city and living conditions of its population. In such a case, the label shift assumption only holds in a slightly modified sense, i.e. $\mathbb{P}(X|Y = y, Z = z) = \mathbb{Q}(X|Y = y, Z = z)$. If the attributes $Z$ are observed, then our framework can readily be used to perform importance weighting.

Furthermore, it is not clear whether the final predictor is in fact "better" or more robust to shifts just because it achieves a better target accuracy than a vanilla unweighted estimator. In fact, there is a reason to believe that under certain shift scenarios, the predictor might learn to use spurious correlations to boost accuracy. Finding a procedure which can both learn a robust model and achieve high accuracies on new target sets remains to be an ongoing challenge. Moreover, the current choice of regularization depends on the number of samples rather than data-driven regularization which is more desirable.

An important direction towards active learning for the same disease-symptoms scenario is when we also have an expert for diagnosing a limited number of patients in the target location. Now the question is which patients would be most "useful" to diagnose to obtain a high accuracy on the entire target set? Furthermore, in the case of high risk, we might be able to choose some of the patients for further medical diagnosis or treatment, up to some varying cost. We plan to extend the current framework to the active learning setting where we actively query the label of certain $x$'s (Beygelzimer et al., 2009) as well as the cost-sensitive setting where we also consider the cost of querying labels (Krishnamurthy et al., 2017).

Consider a realizable and over-parameterized setting, where there exists a deterministic mapping from $x$ to $y$, and also suppose a perfect interpolation of the source data with a minimum proper norm is desired. In this case, weighting the samples in the empirical loss might not alter the trained classifier (Belkin et al., 2018). Therefore, our results might not directly help the design of better classifiers in this particular regime. However, for the general overparameterized settings, it remains an open problem of how the importance weighting can improve the generalization. We leave this study for future work.

## 6 ACKNOWLEDGEMENT

K. Azizzadenesheli is supported in part by NSF Career Award CCF-1254106. This research has been conducted when the first author was a visiting researcher at Caltech. Anqi Liu is supported in part by DOLCIT Postdoctoral Fellowship at Caltech and Caltech's Center for Autonomous Systems and Technologies. Fan Yang is supported by the Institute for Theoretical Studies ETH Zurich and the Dr. Max Rössler and the Walter Haefner Foundation. A. Anandkumar is supported in part by Microsoft Faculty Fellowship, Google faculty award, Adobe grant, NSF Career Award CCF- 1254106, and AFOSR YIP FA9550-15-1-0221.

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

## A MORE EXPERIMENTAL RESULTS

This section contains more experiments that provide more insights about in which settings the advantage of using RLLS vs. BBSL are more or less pronounced.

### A.0.1 CIFAR10 EXPERIMENTS UNDER TWEAK-ONE SHIFT AND DIRICHLET SHIFT

Here we compare weight estimation performance between RLLS and BBSL for different types of shifts including the *Tweak-one Shift*, for which we randomly choose one class, e.g. $i$ and set $p(i) = \rho$ while all other classes are distributed evenly. Figure 5 depicts the the weight estimation performance of RLLS compared to BBSL for a variety of values of $\rho$ and $\alpha$. Note that larger shifts correspond to smaller $\alpha$ and larger $\rho$. In general, one observes that our RLLS estimator has smaller MSE and that as the shift increases, the error of both methods increases. For tweak-one shift we can additionally see that as the shift increases, RLLS outperforms BBSL more and more as both in terms of bias and variance.

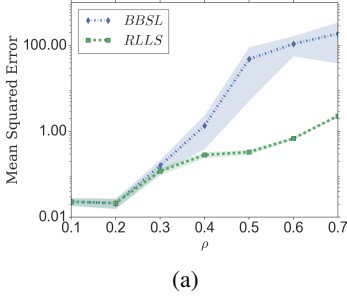
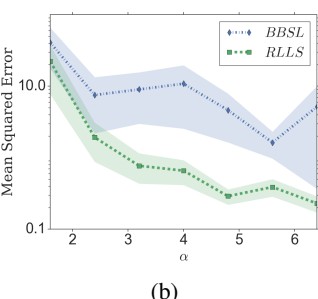

(a)                                       (b)

Figure 5: Comparing MSE of estimated weights using BBSL and RLLS on CIFAR10 with (a) tweak-one shift on source and uniform target, and (b) Dirichlet shift on source and uniform target. $h_0$ is trained using the same source shifted data respectively.

### A.1 MNIST EXPERIMENTS UNDER MINORITY-CLASS SOURCE SHIFTS FOR DIFFERENT VALUES OF $p$

In order to show weight estimation and classification performance under different level of label shifts, we include several additional sets of experiments here in the appendix. Figure 6 shows the weight estimation error and accuracy comparison under a minority-class shift with p = 0.001. The training and testing sample size is 10000 examples in this case. We can see that whenever the weight estimation of RLLS is better, the accuracy is also better, except in the four classes case when both methods are bad in weight estimation.

(a)                                       (b)

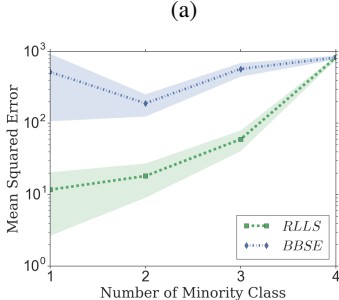
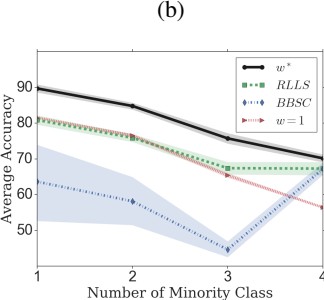

Figure 6: (a) Mean squared error in estimated weights and (b) accuracy on MNIST for minority-class shifted source and uniform target with p = 0.001.

Figure 7 demonstrates another case in minority-class shift when $p = 0.01$. The black-box classifier is the same two-layers neural network trained on a biased source data set with tweak-one $\rho = 0.5$. We observe that when the number of minority class is small like 1 or 2, the weight estimation is similar

between two methods, as well as in the classification accuracy. But when the shift get larger, the weights are worse and the performance in accuracy decreases, getting even worse than the unweighted classifier.

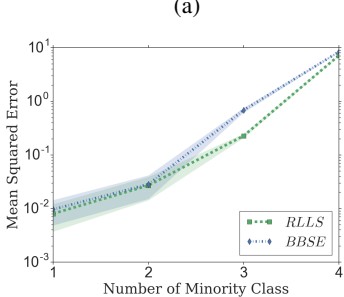
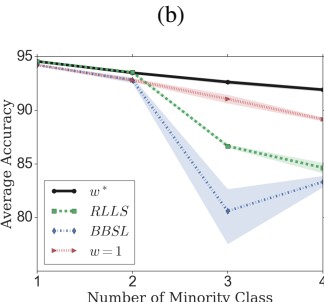

Figure 7: (a) Mean squared error in estimated weights and (b) accuracy on MNIST for minority-class shifted source and uniform target with p = 0.01, with $h_0$ trained on tweak-one shifted source data.

Figure 8 illustrates the weight estimation alongside final classification performance for Minority-Class source shift of MNIST. We use 1000 training and testing data. We created large shifts of three or more minority classes with $p = 0.005$. We use a fixed black-box classifier that is trained on biased source data, with tweak-one $\rho = 0.5$. Observe that the MSE in weight estimation is relatively large and RLLS outperforms BBSL as the number of minority classes increases. As the shift increases the performance for all methods deteriorates. Furthermore, Figure 8 (b) illustrates how the advantage of RLLS over the unweighted classifier increases as the shift increases. Across all shifts, the RLLS based classifier yields higher accuracy than the one based on BBSL.

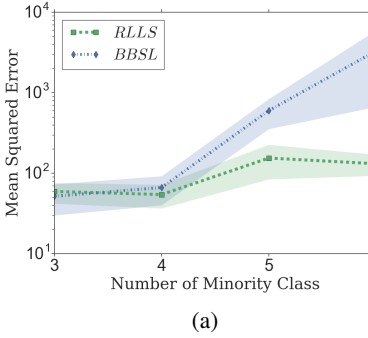
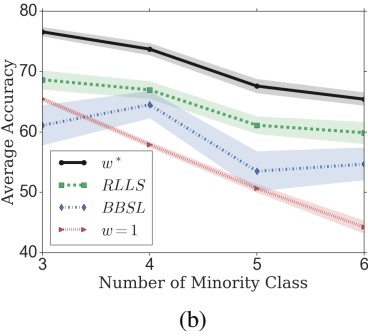

Figure 8: (a) Mean squared error in estimated weights and (b) accuracy on MNIST for minority-class shifted source and uniform target with p = 0.005, with $h_0$ trained on tweak-one shifted source data.

### A.2  CIFAR10 EXPERIMENT UNDER DIRICHLET SOURCE SHIFTS

Figure 9 illustrates the weight estimation alongside final classification performance for Dirichlet source shift of CIFAR10 dataset. We use 10000 training and testing data in this experiment, following the way we generate shift on source data. We train $h_0$ with tweak-one shifted source data with $\rho = 0.5$. The results show that importance weighting in general is not helping the classification in this relatively large shift case, because the weighted methods, including true weights and estimated weights, are similar in accuracy with the unweighted method.

### A.3  MNIST EXPERIMENT UNDER DIRICHLET SHIFT WITH LOW TARGET SAMPLE SIZE

We show the performance of classifier with different regularization $\lambda$ under a Dirichlet shift with $\alpha = 0.5$ in Figure 10. The training has 5000 examples in this case. We can see that in this low target sample case, $\lambda = 1$ only take over after several hundreds example, while some $\lambda$ value between 0 and 1 outperforms it at the beginning. Similar as in the paper, we use different black-box classifier that is corrupted in different levels to show the relation between the quality of black-box predictor and the necessary target sample size. We use biased source data with tweak-one $\rho = 0, 0.2, 0.6$ to train the black-box classifier. We see that we need more target samples for the fully weighted version $\lambda = 1$ to take over for a more corrupted black-box classifier.

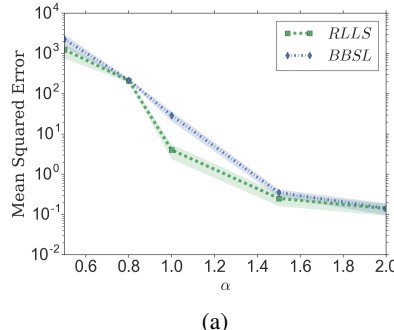 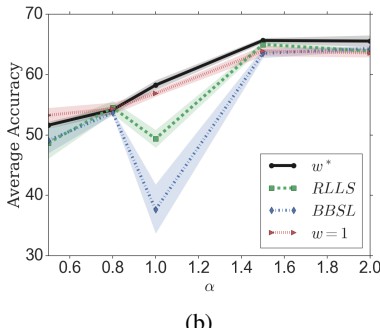

Figure 9: (a) Mean squared error in estimated weights and (b) accuracy on CIFAR10 for Dirichlet shifted source and uniform target. with $h_0$ trained on tweak-one shifted source data.

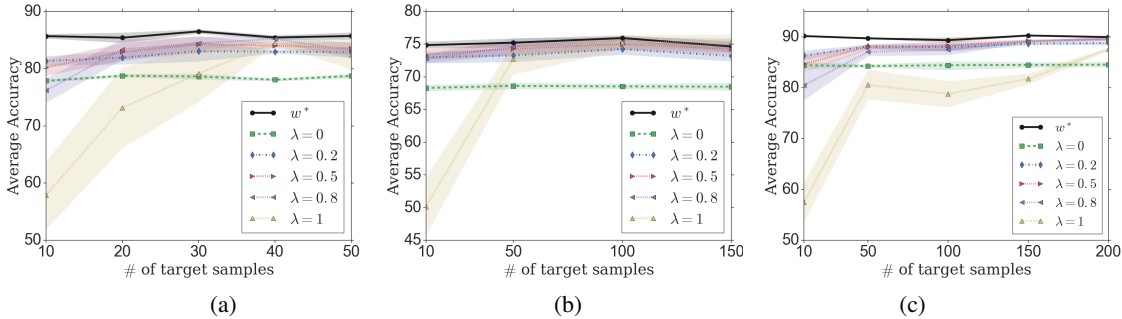

Figure 10: Performance on MNIST for Dirichlet shifted source and uniform target with various target sample size and $\lambda$ using (a) better predictor, (b) neutral predictor and (c) corrupted predictor.

## B PROOFS

### B.1 PROOF OF LEMMA 1

From Thm. 3.4 in (Pires & Szepesvári, 2012) we know that for $\widehat{\theta}$ as defined in equation (3), if with probability at least $1 - \delta$, $\|\widehat{C} - C\|_2 \leq \Delta_C$ and $\|\widehat{b} - b\|_2 \leq \Delta_b$ hold simultaneously, then

$$\Upsilon(\widehat{\theta}) \leq \inf_{\theta' \in \mathcal{R}^k} \{\Upsilon(\theta') + 2\Delta_C \|\theta'\|_\alpha\} + 2\Delta_b. \tag{7}$$

where we use the shorthand $\Upsilon(\theta') = \|C\theta' - b\|_2$.

We can get an upper bound on the right hand side of (7) is the infimum by simply choosing a feasible $\theta' = \theta$. We then have $\|C\theta - b\|_2 = 0$ and hence

$$\inf_{\theta'} \{\Upsilon(\theta') + 2\Delta_C \|\theta'\|_2\} \leq 2\Delta_C \|\theta\|_2$$

as a consequence,

$$\Upsilon(\widehat{\theta}) = \|C\widehat{\theta} - b\|_2 = \|C\left(\widehat{\theta} - \theta\right)\|_2 \leq 2\Delta_C \|\theta\|_2 + 2\Delta_b$$

Since $\|C\left(\widehat{\theta} - \theta\right)\|_2 \geq \sigma_{\min}(C)\|\widehat{\theta} - \theta\|_2$ by definition of the minimum singular value, we thus have

$$\|\widehat{\theta} - \theta\|_2 \leq \frac{1}{\sigma_{min}} \left(2\Delta_C \|\theta\|_2 + 2\Delta_b\right)$$

Let us first notice that

$$b_h = q_h - C\mathbf{1} = q_h - p_h$$

The mathematical definition of the finite sample estimates $\widehat{C}_h, \widehat{b}_h$ (in matrix and vector representation) with respect to some hypothesis $h$ are as follows

$$[\widehat{C}_h]_{ij} = \frac{1}{(1-\beta)n_p} \sum_{(x,y) \in D_p} \mathbb{I}_{h(x)=i, y=j}$$

$$[\widehat{b}_h](i) = \frac{1}{m} \sum_{(x,y) \in D_q} \mathbb{I}_{h(x)=i} - \frac{1}{(1-\beta)n_p} \sum_{(x,y) \in D_p^{\text{weight}}} \mathbb{I}_{h(x)=i}$$

where $m = |D_q|$ and $\mathbb{I}$ is the indicator function. $C_h, b_h$ can equivalently be expressed with the population over $\mathbb{P}$ for $C_h$ and over $\mathbb{Q}$ for $b_h$ respectively. We now use the following concentration Lemmas to bound the estimation errors of $\widehat{C}, \widehat{b}$ where we drop the subscript $h$ for ease of notation.

**Lemma 2 (Concentration of measurement matrix $\widehat{C}$)** *For finite sample estimate $\widehat{C}$ we have*

$$\|\widehat{C} - C\|_2 \leq \frac{2\log(2k/\delta)}{3(1-\beta)n_p} + \sqrt{\frac{2\log(2k/\delta)}{(1-\beta)n_p}}$$

*with probability at least $(1 - \delta)$.*

**Lemma 3 (Concentration of label measurements)** *For the finite sample estimate $\widehat{b}$ with respect to any hypothesis $h$ it holds that*

$$\|\widehat{b}_h - b_h\|_2 \leq \frac{2}{\sqrt{\log(2)}} \left( \frac{\sqrt{\log(1/\delta)}}{\sqrt{(1-\beta)n_p}} + \frac{\sqrt{\log(1/\delta)}}{\sqrt{n_q}} \right)$$

*with probability at least $1 - 2\delta$.*

By Lemma. 2 for concentration of $C$ and Lemma. 3 for concentration of $b$ we now have with probability at least $1 - \delta$

$$\|\widehat{\theta} - \theta\|_2 \leq \frac{1}{\sigma_{min}} \left( \frac{2\|\theta\|_2 \log(2k/\delta)}{(1-\beta)n_p} + \|\theta\|_2 \sqrt{\frac{18\log(4k/\delta)}{(1-\beta)n_p}} \right.$$
$$\left. + \sqrt{\frac{36\log(2/\delta)}{n_q}} + \sqrt{\frac{36\log(2/\delta)}{(1-\beta)n_p}} \right).$$

which, considering that $O(\frac{1}{\sqrt{n}})$ dominates $O(\frac{1}{n})$, yields the statement of the Lemma 1.

## B.2 PROOF OF LEMMA 2

We prove this lemma using the theorem 1.4[Matrix Bernstein] and Dilations technique from Tropp (2012). We can rewrite $C_h = \mathbb{E}_{(x,y) \sim \mathbb{P}} \left[ e_{h(x)} e_y^\top \right]$ where $e(i)$ is the one-hot-encoding of index $i$. Consider a finite sequence $\{\Psi(i)\}$ of independent random matrices with dimension $k$. By dilations, lets construct another sequence of self-adjoint random matrices of $\{\widetilde{\Psi}(i)\}$ of dimension $2k$, such that for all $i$

$$\widetilde{\Psi}(i) = \begin{bmatrix} \mathbf{0} & \Psi(i) \\ \Psi(i)^\top & \mathbf{0} \end{bmatrix}$$

therefore,

$$\widetilde{\Psi}(i)^2 = \begin{bmatrix} \Psi(i)\Psi(i)^\top & \mathbf{0} \\ \mathbf{0} & \Psi(i)^\top\Psi(i) \end{bmatrix} \tag{8}$$

which results in $\|\widetilde{\Psi}(i)\|_2 = \|\Psi(i)\|_2$. The dilation technique translates the initial sequence of random matrices to the sequence of random self-adjoint matrices where we can apply the Matrix Bernstein theorem which states that, for a finite sequence of i.i.d. self-adjoint matrices $\widetilde{\Psi}(i)$, such that, almost surely $\forall i$, $\mathbb{E}\left[\widetilde{\Psi}(i)\right] = 0$ and $\|\widetilde{\Psi}(i)\| \leq R$, then for all $t \geq 0$,

$$\|\frac{1}{t}\sum_{i=1}^{t} \widetilde{\Psi}(i)\| \leq \frac{R\log(2k/\delta)}{3t} + \sqrt{\frac{2\varrho^2 \log(2k/\delta)}{t}}$$

with probability at least $1 - \delta$ where $\varrho^2 := \|\mathbb{E}\left[\widetilde{\Psi}^2(i)\right]\|_2$, $\forall i$ which is also $\varrho^2 = \|\mathbb{E}\left[\Psi^2(i)\right]\|_2$, $\forall i$ due to Eq. 8. Therefore, thanks to the dilation trick and theorem 1.4[Matrix Bernstein] in Tropp (2012),

$$\|\frac{1}{t}\sum_{i=1}^{t}\Psi(i)\| \leq \frac{R\log\left(2k/\delta\right)}{3t} + \sqrt{\frac{2\varrho^2\log\left(2k/\delta\right)}{t}}$$

with probability at least $1 - \delta$.

Now, by plugging in $\Psi(i) = e_{h(x(i))}e_{y(i)}^\top - C$, we have $\mathbb{E}\left[\widetilde{\Psi}(i)\right] = 0$. Together with $\|\widetilde{\Psi}(i)\| \leq 2$ as well as $\varrho^2 = \|\mathbb{E}\left[\Psi^2(i)\right]\|_2 = 1$ and setting $t = n$, we have

$$\|\widehat{C} - C\|_2 \leq \frac{2\log\left(2k/\delta\right)}{3(1-\beta)n} + \sqrt{\frac{2\log\left(2k/\delta\right)}{(1-\beta)n}}$$

### B.3 PROOF OF LEMMA 3

The proof of this lemma is mainly based on a special case of and appreared at proposition 6 in Azizzadenesheli et al. (2016), Lemma F.1 in Anandkumar et al. (2012) and Proposition 19 of Hsu et al. (2012).

Analogous to the previous section we can rewrite $b_h = \mathbb{E}_{(x,y)\in\mathbb{Q}}[e_{h(x)}] - \mathbb{E}_{(x,y)\in\mathbb{P}}[e_{h(x)}]$ where $e(i)$ is the one-hot-encoding of index $i$. Note that (dropping the subscript $h$) we have

$$\|\widehat{b}_h - b_h\|_2 \leq \|\widehat{q}_h - q_h\|_2 + \|\widehat{p}_h - p_h\|_2$$

We now bound both estimates of probability vectors separately.

Consider a fixed multinomial distribution characterized with probability vector of $\overline{\varsigma} \in \Delta_{k-1}$ where $\Delta_{k-1}$ is a $k - 1$ dimensional simplex. Further, consider $t$ realization of this multinomial distribution $\{\varsigma(i)\}_{i=1}^{t}$ where $\varsigma(i)$ is the one-hot-encoding of the $i$'th sample. Consider the empirical estimate mean of this distribution through empirical average of the samples; $\widehat{\varsigma} = \frac{1}{t}\sum(i)^t\varsigma(i)$, then

$$\|\widehat{\varsigma} - \overline{\varsigma}\| \leq \frac{1}{\sqrt{t}} + \sqrt{\frac{\log\left(1/\delta\right)}{t}}$$

with probability at least $1 - \delta$.

By plugging in $\overline{\varsigma} = q_h, \widehat{\varsigma} = \widehat{q}_h$ with $t = n_q$ and finally $\{\varsigma(i)\}_{i=1}^{n_q} = \{e_{h(x(i))}\}(i)^{n_q}$ and equivalently for $p_h$ we obtain;

$$\|\widehat{b}_h - b_h\|_2 \leq \left(\frac{\sqrt{\log(1/\delta)}}{\sqrt{(1-\beta)n_p}} + \frac{\sqrt{\log(1/\delta)}}{\sqrt{n_q}}\right) + \left(\frac{1}{\sqrt{(1-\beta)n_p}} + \frac{1}{\sqrt{n_q}}\right)$$

with probability at least $1 - 2\delta$, therefore;

$$\|\widehat{b}_h - b_h\|_2 \leq \frac{2}{\sqrt{\log(2)}}\left(\frac{\sqrt{\log(1/\delta)}}{\sqrt{(1-\beta)n_p}} + \frac{\sqrt{\log(1/\delta)}}{\sqrt{n_q}}\right)$$

resulting in the statement in the Lemma 3.

### B.4 PROOF OF THEOREM 1

We want to ultimately bound $|\mathcal{L}(\widehat{h}_{\widehat{w}}) - \mathcal{L}(h^\star)|$. By addition and subtraction we have

$$\mathcal{L}(\widehat{h}_{\widehat{w}}) - \mathcal{L}(h^\star) = \underbrace{\mathcal{L}(\widehat{h}_{\widehat{w}}) - \mathcal{L}_n(\widehat{h}_{\widehat{w}})}_{(b)} + \underbrace{\mathcal{L}_n(\widehat{h}_{\widehat{w}}) - \mathcal{L}_n(\widehat{h}_{\widehat{w}};\widehat{w})}_{(a)}$$
$$+ \underbrace{\mathcal{L}_n(\widehat{h}_{\widehat{w}};\widehat{w}) - \mathcal{L}_n(h^\star;\widehat{w})}_{\leq 0} + \underbrace{\mathcal{L}_n(h^\star;\widehat{w}) - \mathcal{L}_n(h^\star)}_{(a)} + \underbrace{\mathcal{L}_n(h^\star) - \mathcal{L}(h^\star)}_{(b)} \quad (9)$$

where $n = \beta n_p$ and we used optimality of $\widehat{h}_{\widehat{w}}$. Here (a) is the weight estimation error and (b) is the finite sample error.

**Uniform law for bounding (b)**   We bound (b) using standard results for uniform laws for uniformly bounded functions which holds since $\|w\|_\infty \le d_\infty(q||p)$ and $\|\ell\|_\infty \le 1$. Since $|w(y)\ell(h(x), y)| \le d_\infty(q||p)$, $\forall x, y \in \mathcal{X} \times \mathcal{Y}$, by deploying the McDiarmid's inequality we then obtain that

$$\sup_{h \in \mathcal{H}} |\mathcal{L}_n(h) - \mathcal{L}(h)| \le 2\mathcal{R}_n(\mathcal{G}(\ell, \mathcal{H})) + d_\infty(q||p)\sqrt{\frac{\log \frac{2}{\delta}}{n}}$$

where $\mathcal{G}(\ell, \mathcal{H}) = \{g_h(x, y) = w(y)\ell(h(x), y) : h \in \mathcal{H}\}$ and the Rademacher complexity is defined as $\mathcal{R}_n(\mathcal{G}) := \mathbb{E}_{X(i), Y(i) \sim \mathbb{P}: i \in [n]} \left[\mathbb{E}_{\sigma_i : i \in [n]} \frac{1}{n} [\sup_{h \in \mathcal{H}} \sum_{i=1}^n \sigma_i w(y_i)\ell(x_i, h(y_i))]\right]$.

of the hypothesis class $\mathcal{H}$ (see for example Percy Liang notes on Statistical Learning Theory and chapter 4 in Wainwright (2019))

**Bounding term (a)**   Remember that $k = |\mathcal{Y}|$ is the cardinality of the finite domain of $Y$, or the number of classes. Let us define $\widetilde{\ell} \in \mathcal{R}^k$ with $\widetilde{\ell}_j = \sum_{i=1}^n \mathbb{I}_{y(i)=j}\ell(y(i), h(x(i)))$. Notice that by definition $\|\widetilde{\ell}\|_1 \le n$ and $\|\widetilde{\ell}\|_\infty \le n$ from which it follows by Hoelder's inequality that $\|\widetilde{\ell}\|_2 \le n$. Furthermore, we slightly abuse notation and use $w$ to denote the $k$-dimensional vector with $w_i = w(i)$. Therefore, for all $h$ we have via the Cauchy Schwarz inequality that

$$\begin{aligned}
|\mathcal{L}_n(h; w) - \mathcal{L}_n(h; \widehat{w})| &= |\frac{1}{n}\sum_{i=1}^n (w(y(i)) - \widehat{w}(y(i)))\ell(h(x(i)), y(i))| \\
&\le |\frac{1}{n}\sum_{j=1}^k (w(j) - \widehat{w}(j))\widetilde{\ell}(j)| \\
&\le \frac{1}{n}\|\widehat{w} - w\|_2\|\widetilde{\ell}\|_2 \le \|\widehat{w} - w\|_2 \\
&\le \|\lambda\widehat{\theta} - \theta\|_2 \le (1-\lambda)\|\theta\|_2 + \lambda\|\widehat{\theta} - \theta\|_2
\end{aligned} \tag{10}$$

It then follows by Lemma 1 that

$$\sup_{h \in \mathcal{H}} |\mathcal{L}_n(h; w) - \mathcal{L}_n(h; \widehat{w})| \le (1-\lambda)\|\theta\|_2$$

$$O\left(\frac{\lambda}{\sigma_{min}}\left((\|\theta\|_2)\sqrt{\frac{\log(k/\delta)}{(1-\beta)n_p}} + \sqrt{\frac{\log(1/\delta)}{(1-\beta)n_p}}\sqrt{\frac{\log(1/\delta)}{n_q}}\right)\right)$$

**Lemma 4 (McDiarmid-Doob-Freedman-Rademacher)** *For a given A hypothesis class $\mathcal{H}$, a set $\mathcal{G}(\ell, \mathcal{H}) = \{g_h(x, y) = w(y)\ell(h(x), y) : h \in \mathcal{H}\}$, under $n$ data points and loss function $\ell$ we have*

$$\sup_{h \in \mathcal{H}} |\mathcal{L}_n(h) - \mathcal{L}(h)| \le 2\mathcal{R}(\mathcal{G}(\ell, \mathcal{H})) + \frac{2d_\infty(q||p)\log(2/\delta)}{n} + \sqrt{2\frac{d(q||p)\log(2/\delta)}{n}}$$

*with probability at least $1 - \delta$*

Plugging both bounds back into equation (9) concludes the proof of the theorem.

### B.5   PROOF OF LEMMA 4

With a bit abuse of notation let's restate the empirical loss with known importance weights instead on the random variables $\{(X_i, Y_i)\}_1^n$

$$\mathcal{L}_n(h) = \frac{1}{n}\sum_{j=1}^n w(Y_j)\ell(Y_j, h(X_j))$$

We further define a ghost data set $\{(X_i', Y_i')\}_1^n$ and the corresponding ghost loss;

$$\mathcal{L}_n'(h) = \frac{1}{n}\sum_{j=1}^n w(Y_j')\ell(Y_j', h(X_j'))$$

Let's define a random variable $G_n := \sup_{h \in \mathcal{H}} \mathcal{L}_n(h) - \mathcal{L}(h)$. This random variable is the key to derive the tight generalization bound in Lemma 4.

This random variable has the following properties;

$$\mathbb{E}\left[G_n\right] = \mathbb{E}\left[\sup_{h \in \mathcal{H}} \mathcal{L}_n(h) - \mathbb{E}\left[\mathcal{L}'_n(h)\right]\right]$$

Which we can rewrite as

$$\mathbb{E}\left[G_n\right] = \mathbb{E}\left[\sup_{h \in \mathcal{H}} \mathbb{E}\left[\mathcal{L}_n(h) - \mathcal{L}'_n(h)\Big|\{(X_i, Y_i)\}_1^n\right]\right]$$

and swapping the sup with the expectation

$$\mathbb{E}\left[G_n\right] \leq \mathbb{E}\left[\mathbb{E}\left[\sup_{h \in \mathcal{H}} \mathcal{L}_n(h) - \mathcal{L}'_n(h)\Big|\{(X_i, Y_i)\}_1^n\right]\right]$$

We can remove the condition with law of iterated conditional expectation and have expectation on both of the data sets;

$$\mathbb{E}\left[G_n\right] \leq \mathbb{E}\left[\sup_{h \in \mathcal{H}} \mathcal{L}_n(h) - \mathcal{L}'_n(h)\right]$$

we further open the expression up;

$$\mathbb{E}\left[G_n\right] \leq \mathbb{E}\left[\sup_{h \in \mathcal{H}} \frac{1}{n}\sum_i^n w_i \ell(h(X_i), Y_i) - w'_i \ell'(h(X'_i), Y'_i)\right]$$

In the following we use the usual symmetrizing technique through Rademacher variables $\{\xi_i\}_1^n$. Each $\xi_i$ is a uniform random variable either 1 or $-1$. Therefore since $\ell(h(X'), Y') - \ell'(h(X'), Y')$ is a symmetric random variable we have

$$\mathbb{E}\left[G_n\right] \leq \mathbb{E}\left[\sup_{h \in \mathcal{H}} \frac{1}{n}\sum_i^n \xi_i \left[w(Y_i)\ell(h(X_i), Y_i) - w(Y'_i)\ell'(h(X'_i), Y'_i)\right]\right]$$

where the expectation is also over the Rademacher variables. After propagation sup

$$\mathbb{E}\left[G_n\right] \leq \mathbb{E}\left[\sup_{h \in \mathcal{H}} \frac{1}{n}\sum_i^n \xi_i w(Y_i)\ell(h(X_i), Y_i) + \sup_{h \in \mathcal{H}} \frac{1}{n}\sum_i^n -\xi_i w(Y'_i)\ell'(h(X'_i), Y'_i)\right]$$

By propagating the expectation and again symmetry in the Rademacher variable we have

$$\mathbb{E}\left[G_n\right] \leq 2\mathbb{E}\left[\sup_{h \in \mathcal{H}} \frac{1}{n}\sum_i^n \xi_i w(Y_i)\ell(h(X_i), Y_i)\right] = 2\mathcal{R}(\mathcal{G}(\ell, \mathcal{H}))$$

where the right hand side is two times the Rademacher complexity of class $\mathcal{G}(\ell, \mathcal{H})$. Consider a sequence of Doob Martingale and filtration $(U_j, \mathcal{F}_j)$ defined on some probability space $(\Omega, \mathcal{F}, Pr)$;

$$U_j := \mathbb{E}\left[\sup_{h \in \mathcal{H}} \frac{1}{n}\sum_i^n w(Y_i)\ell(h(X_i), Y_i)\Big|\{(X_i, Y_i)\}_1^j\right]$$

and the corresponding Martingale difference;

$$D_j := U_j - U_{j-1}$$

In the following we show that each $|D_j|$ is bounded above.

$$D_j = \mathbb{E}\left[\sup_{h \in \mathcal{H}} \frac{1}{n}\sum_i^n w(Y_i)\ell(h(X_i), Y_i)\Big|\{(X_i, Y_i)\}_1^j\right] - \mathbb{E}\left[\sup_{h \in \mathcal{H}} \frac{1}{n}\sum_i^n w(Y_i)\ell(h(X_i), Y_i)\Big|\{(X_i, Y_i)\}_1^{j-1}\right]$$

$$\leq \max_{x_j, y_j} \mathbb{E}\left[\sup_{h \in \mathcal{H}} \frac{1}{n}\sum_i^n w(Y_i)\ell(h(X_i), Y_i)\Big|\{(X_i, Y_i)\}_1^{j-1}, x_j, y_j\right]$$

$$- \min_{x_j, y_j} \mathbb{E}\left[\sup_{h \in \mathcal{H}} \frac{1}{n}\sum_i^n w(Y_i)\ell(h(X_i), Y_i)\Big|\{(X_i, Y_i)\}_1^{j-1}, x_j, y_j\right]$$

Let's define $x_j^{\max}, y_j^{\max}$ as the solution to the maximization and $x_j^{\min}, y_j^{\min}$ the solution to the minimization, therefore,

$$
\begin{aligned}
\mathcal{D}_j &\leq \mathbb{E}\left[\sup_{h\in\mathcal{H}} \frac{1}{n}\sum_i^n w(Y_i)\ell(h(X_i),Y_i)\Big|\{(X_i,Y_i)\}_1^{j-1}, x_j^{\max}, y_j^{\max}\right] \\
&\qquad - \mathbb{E}\left[\sup_{h\in\mathcal{H}} \frac{1}{n}\sum_i^n w(Y_i)\ell(h(X_i),Y_i)\Big|\{(X_i,Y_i)\}_1^{j-1}, x_j^{\min}, y_j^{\min}\right] \\
&= \mathbb{E}\left[\sup_{h\in\mathcal{H}} \left(\frac{1}{n}\sum_i^n w(Y_i)\ell(h(X_i),Y_i) + w(y_j^{\min})\ell(h(x_j^{\min}),y_j^{\min}) - w(y_j^{\min})\ell(h(x_j^{\min}),y_j^{\min})\right)\Big|\{(x_i,y_i)\}_1^{j-1}, x_j^{\max}, y_j^{\max}\right] \\
&\qquad - \mathbb{E}\left[\sup_{h\in\mathcal{H}} \frac{1}{n}\sum_i^n w(Y_i)\ell(h(X_i),Y_i)\Big|\{(X_i,Y_i)\}_1^{j-1}, x_j^{\min}, y_j^{\min}\right] \\
&= \mathbb{E}\left[\sup_{h\in\mathcal{H}} \left(\frac{1}{n}\sum_i^n w(Y_i)\ell(h(X_i),Y_i) + w(y_j^{\max})\ell(h(x_j^{\max}),y_j^{\max}) - w(y_j^{\min})\ell(h(x_j^{\min}),y_j^{\min})\right)\Big|\{(x_i,y_i)\}_1^{j-1}, x_j^{\min}, y_j^{\min}\right] \\
&\qquad - \mathbb{E}\left[\sup_{h\in\mathcal{H}} \frac{1}{n}\sum_i^n w(Y_i)\ell(h(X_i),Y_i)\Big|\{(X_i,Y_i)\}_1^{j-1}, x_j^{\min}, y_j^{\min}\right] \\
&\leq \frac{1}{n}\sup_{h\in\mathcal{H}} \left|w(y_j^{\max})\ell(h(x_j^{\max}),y_j^{\max}) - w(y_j^{\min})\ell(h(x_j^{\min}),y_j^{\min})\right| \leq \frac{d_\infty(q||p)}{n}
\end{aligned}
$$

The same way we can bound $-D_j$. Therefore the absolute value each $D_j$ is bounded by $\frac{d_\infty(q||p)}{n}$. In the following we bound the conditional second moment, $\mathbb{E}\left[D_j^2|\mathcal{F}_{j-1}\right]$;

$$
\mathbb{E}\left[\left(\underbrace{\mathbb{E}\left[\sup_{h\in\mathcal{H}} \frac{1}{n}\sum_i^n w(Y_i)\ell(h(X_i),Y_i)\Big|\{(X_i,Y_i)\}_1^{j}\right]}_{(a')} - \underbrace{\mathbb{E}\left[\sup_{h\in\mathcal{H}} \frac{1}{n}\sum_i^n w(Y_i)\ell(h(X_i),Y_i)\Big|\{(X_i,Y_i)\}_1^{j-1}\right]}_{(b')}\right)^2 \Big|\{(X_i,Y_i)\}_1^{j-1}\right]
$$

Let's construct an event $\mathcal{C}_j$ the event that $a'$ is bigger than $b'$, and also $\mathcal{C}_j'$ its compliment. Therefore, for the $\mathbb{E}\left[D_j^2|\mathcal{F}_{j-1}\right]$ we have

$$
\begin{aligned}
&\mathbb{E}\left[\left(\mathbb{E}\left[\sup_{h\in\mathcal{H}} \frac{1}{n}\sum_i^n w(Y_i)\ell(h(X_i),Y_i)\Big|\{(X_i,Y_i)\}_1^{j}\right]\right.\right. \\
&\qquad\qquad \left.\left. - \mathbb{E}\left[\sup_{h\in\mathcal{H}} \frac{1}{n}\sum_i^n w(Y_i)\ell(h(X_i),Y_i)\Big|\{(X_i,Y_i)\}_1^{j-1}\right]\right)^2 \Big|\{(X_i,Y_i)\}_1^{j-1}, C_j\right]\mathbb{P}(C_j|\mathcal{F}_{j-1}) \\
&+ \mathbb{E}\left[\left(\mathbb{E}\left[\sup_{h\in\mathcal{H}} \frac{1}{n}\sum_i^n w(Y_i)\ell(h(X_i),Y_i)\Big|\{(X_i,Y_i)\}_1^{j}\right]\right.\right. \\
&\qquad\qquad \left.\left. - \mathbb{E}\left[\sup_{h\in\mathcal{H}} \frac{1}{n}\sum_i^n w(Y_i)\ell(h(X_i),Y_i)\Big|\{(X_i,Y_i)\}_1^{j-1}\right]\right)^2 \Big|\{(X_i,Y_i)\}_1^{j-1}, C_j'\right]\mathbb{P}(C_j'|\mathcal{F}_{j-1})
\end{aligned}
$$

$$(11)$$

For the firs term in Eq. 11 after again introducing ghost variables $X', Y'$ we have the following upper bound

$$
\mathbb{E}\Bigg[\Bigg(\mathbb{E}\bigg[\sup_{h\in\mathcal{H}}\frac{1}{n}\sum_{i\neq j}^{n}w(Y_i)\ell(h(X_i),Y_i)+\frac{1}{n}\sup_{h\in\mathcal{H}}w(Y_j)\ell(h(X_j),Y_j)\Big|\{(X_i,Y_i)\}_1^j\bigg]
$$
$$
-\mathbb{E}\bigg[\sup_{h\in\mathcal{H}}\frac{1}{n}\sum_{i}^{n}w(Y_i)\ell(h(X_i),Y_i)\Big|\{(X_i,Y_i)\}_1^{j-1}\bigg]\Bigg)^2\Big|\{(X_i,Y_i)\}_1^{j-1},C_j\Bigg]\mathbb{P}(C_j|\mathcal{F}_{j-1})
$$
$$
\leq\mathbb{E}\Bigg[\Bigg(\mathbb{E}\bigg[\sup_{h\in\mathcal{H}}\frac{1}{n}\sum_{i\neq j}^{n}w(Y_i)\ell(h(X_i),Y_i)+\frac{1}{n}w(Y_j')\ell(h(X_j'),Y_j')+\frac{1}{n}\sup_{h\in\mathcal{H}}w(Y_j)\ell(h(X_j),Y_j)\Big|\{(X_i,Y_i)\}_1^j\bigg]
$$
$$
-\mathbb{E}\bigg[\sup_{h\in\mathcal{H}}\frac{1}{n}\sum_{i}^{n}w(Y_i)\ell(h(X_i),Y_i)\Big|\{(X_i,Y_i)\}_1^{j-1}\bigg]\Bigg)^2\Big|\{(X_i,Y_i)\}_1^{j-1},C_j\Bigg]\mathbb{P}(C_j|\mathcal{F}_{j-1})
$$
$$
\leq\mathbb{E}\Bigg[\Bigg(\mathbb{E}\bigg[\sup_{h\in\mathcal{H}}\frac{1}{n}\sum_{i}^{n}w(Y_i)\ell(h(X_i),Y_i)\Big|\{(X_i,Y_i)\}_1^{j-1}\bigg]+\frac{1}{n}\sup_{h\in\mathcal{H}}w(Y_j)\ell(h(X_j),Y_j)
$$
$$
-\mathbb{E}\bigg[\sup_{h\in\mathcal{H}}\frac{1}{n}\sum_{i}^{n}w(Y_i)\ell(h(X_i),Y_i)\Big|\{(X_i,Y_i)\}_1^{j-1}\bigg]\Bigg)^2\Big|\{(X_i,Y_i)\}_1^{j-1},C_j\Bigg]\mathbb{P}(C_j|\mathcal{F}_{j-1})
$$
$$
\leq\mathbb{E}\Bigg[\Bigg(\frac{1}{n}\sup_{h\in\mathcal{H}}w(Y_j)\ell(h(X_j),Y_j)\Bigg)^2\Big|\{(X_i,Y_i)\}_1^{j-1},C_j\Bigg]\mathbb{P}(C_j|\mathcal{F}_{j-1})
$$
$$
\leq\mathbb{E}\Bigg[\Bigg(\frac{1}{n}\sup_{h\in\mathcal{H}}w(Y_j)\ell(h(X_j),Y_j)\Bigg)^2\Big|\{(X_i,Y_i)\}_1^{j-1}\Bigg]\mathbb{P}(C_j|\mathcal{F}_{j-1})\leq\frac{d(q||p)}{n^2}\mathbb{P}(C_j|\mathcal{F}_{j-1})
$$

So far we have that the first term in Eq. 11 is bounded by $\frac{d(q||p)}{n^2}$. Now for the second term we have the following upper bound;

$$
\mathbb{E}\Bigg[\Bigg(\mathbb{E}\bigg[\sup_{h\in\mathcal{H}}\frac{1}{n}\sum_{i}^{n}w(Y_i)\ell(h(X_i),Y_i)\Big|\{(X_i,Y_i)\}_1^{j-1}\bigg]
$$
$$
-\mathbb{E}\bigg[\sup_{h\in\mathcal{H}}\frac{1}{n}\sum_{i}^{n}w(Y_i)\ell(h(X_i),Y_i)\Big|\{(X_i,Y_i)\}_1^{j}\bigg]\Bigg)^2\Big|\{(X_i,Y_i)\}_1^{j-1},C_j'\Bigg]\mathbb{P}(C_j'|\mathcal{F}_{j-1})
$$
$$
\leq\mathbb{E}\Bigg[\Bigg(\mathbb{E}\bigg[\sup_{h\in\mathcal{H}}\frac{1}{n}\sum_{i\neq j}^{n}w(Y_i)\ell(h(X_i),Y_i)+\frac{1}{n}\sup_{h\in\mathcal{H}}w(Y_j)\ell(h(X_j),Y_j)\Big|\{(X_i,Y_i)\}_1^{j-1}\bigg]
$$
$$
-\mathbb{E}\bigg[\sup_{h\in\mathcal{H}}\frac{1}{n}\sum_{i\neq j}^{n}w(Y_i)\ell(h(X_i),Y_i)\Big|\{(X_i,Y_i)\}_1^{j-1}\bigg]\Bigg)^2\Big|\{(X_i,Y_i)\}_1^{j-1},C_j'\Bigg]\mathbb{P}(C_j'|\mathcal{F}_{j-1})
$$
$$
\leq\Bigg(\mathbb{E}\bigg[\frac{1}{n}\sup_{h\in\mathcal{H}}w(Y_j)\ell(h(X_j),Y_j)\Big|\{(X_i,Y_i)\}_1^{j-1}\bigg]\Bigg)^2
$$
$$
\leq\Bigg(\mathbb{E}\bigg[\frac{1}{n}w(Y_j)\Big|\{(X_i,Y_i)\}_1^{j-1}\bigg]\Bigg)^2=\frac{1}{n^2}\mathbb{P}(C_j'|\mathcal{F}_{j-1})
$$

Therefore, since $d(q||p)\geq 1$

$$
\mathbb{E}\left[D_j^2|\mathcal{F}_{j-1}\right]\leq\frac{d(q||p)}{n^2}\mathbb{P}(C_j|\mathcal{F}_{j-1})+\frac{1}{n^2}\mathbb{P}(C_j'|\mathcal{F}_{j-1})\leq\frac{d(q||p)}{n^2}
$$

For the first inequality, we used the fact that the loss is within $[0,1]$ and the second one is from Eq. 12. Since the first term in RHS is $(b')$, therefore, second moment of each $D_j|\mathcal{F}_{j-1}$ is bounded by $\frac{2d(q||p)}{n}$.

Therefore for the Doob Martingale sequence of $D_j$ we have $|D_j| \leq \frac{d_\infty(q||p)}{n}$ as well as $\sum_j^n D_j^2 |\mathcal{F}_{j-1} \leq \frac{d(q||p)}{n}$. Using the Freedman's inequality Freedman (1975), we have

$$P\left(\sum_j^n D_j = G_n - \mathbb{E}\left[G_n\right] \geq \alpha\right) \leq \exp\left(-\frac{\alpha^2}{2(\frac{d(q||p)}{n} + \alpha\frac{d_\infty(q||p)}{n})}\right)$$

Moreover, if we multiply each loss with $-1$, it results in hypothesis class of $-\mathcal{H}$ which has the same Rademacher complexity as $\mathcal{H}$, due the symmetric Rademacher random variable. Let $\widetilde{G}_n$ denote the same quantity as $\widetilde{G}_n$ but on $-\mathcal{H}$. We use this slack variable in order to bound the absolute value of $G_n$. Therefore

$$P\left(G_n \geq \mathbb{E}\left[G_n\right] + \epsilon\right) \leq \exp\left(-\frac{\epsilon^2}{2(\frac{d(q||p)}{n} + \epsilon\frac{d_\infty(q||p)}{n})}\right)$$

$$\leq \exp\left(-\frac{\epsilon^2}{2(\frac{d(q||p)}{n} + \epsilon\frac{d_\infty(q||p)}{n})}\right) \leq \delta/2$$

and the same bound for $\widetilde{G}_n$. By solving it for $\epsilon$ and $\delta$ we have

$$|G_n| \leq 2\mathcal{R}(\mathcal{G}(\ell, \mathcal{H})) + \frac{2d_\infty(q||p)\log(2/\delta)}{n} + \sqrt{2\frac{d(q||p)\log(2/\delta)}{n}}$$

**Note:** A few days prior to the camera ready submission, we realized that a quite similar analysis and statement to Theorem 4 is also studied in Ying (2004).

## B.6 GENERALIZATION FOR FINITE HYPOTHESIS CLASSES

For finite hypothesis classes, one may bound (b) in (9) using Bernstein's inequality.

We bound (b) by first noting that $w(Y)\ell(Y, h(X))$ satisfies the Bernstein conditions so that Bernstein's inequality holds

$$\mathbb{E}_{\mathbb{P}}[w(Y)] = 1, \quad \mathbb{E}_{\mathbb{P}}[w(Y)^2] = d(q||p), \quad \sigma_{\mathbb{P}}^2(w(Y)) = d(q||p) - 1 \tag{12}$$

by definition. Because we assume $\ell \leq 1$, we directly have

$$\mathbb{E}_{\mathbb{P}}\left[w(Y)^2 l^2(Y, h(X))\right] \leq \mathbb{E}_{\mathbb{P}}\left[w(Y)^2\right] = d(q||p) \tag{13}$$

Since we have a bound on the second moment of weighted loss while its first moment is $L(h)$ we can apply Bernstein's inequality to obtain for any fixed $h$ that

$$|\mathcal{L}_n(h) - \mathcal{L}(h)| \leq \frac{2d_\infty(q||p)\log(\frac{2}{\delta})}{3n} + \sqrt{\frac{2\left(d(q||p) - \mathcal{L}(h)^2\right)\log(\frac{2}{\delta})}{n}}$$

For the uniform law for finite hypothesis classes make the union bound on all the hypotheses;

$$\sup_{h \in \mathcal{H}} |\mathcal{L}_n(h) - \mathcal{L}(h)| \leq \frac{2d_\infty(q||p)\log(\frac{2|\mathcal{H}|}{\delta})}{3n} + \sqrt{\frac{2\left(d(q||p)\right)\log(\frac{2|\mathcal{H}|}{\delta})}{n}}$$

The second moment of the importance weighted loss $\mathbb{E}_{\mathbb{P}}\left[\omega(Y)^2\ell^2(h(X), Y)\right]$, given a $h \in \mathcal{H}'$ can be bounded for general $\alpha \geq 0$, potentially leading to smaller values than $d(q||p)$:

$$\mathbb{E}_{\mathbb{P}}\left[\omega_Y^2 \ell^2(Y, h(X))\right]$$

$$= \sum_i^k p(i) \frac{q^2(i)}{p^2(i)} \mathbb{E}_{X \sim p(X|Y=i)}\left[\ell^2(h(X), X)\right]$$

$$= \sum_i^k q(i)^{\frac{1}{\alpha}} \frac{q(i)}{p(i)} q(i)^{\frac{\alpha-1}{\alpha}} \mathbb{E}_{X \sim p(X|Y=i)}\left[\ell^2(h(X), i)\right]$$

$$\leq \left(\sum_i^k q(i) \frac{q(i)^\alpha}{p(i)^\alpha}\right)^{\frac{1}{\alpha}} \left(\sum_i^k q(i) \mathbb{E}_{X \sim p(X|Y=i)}\left[\ell^2(h(X), i)\right]^{\frac{2\alpha}{\alpha-1}}\right)^{\frac{\alpha-1}{\alpha}}$$

$$= \left(\sum_i^k q(i) \frac{q(i)^\alpha}{p(i)^\alpha}\right)^{\frac{1}{\alpha}} \left(\sum_i^k q(i) \mathbb{E}_{X \sim p(X|Y=i)}\left[l^2(h(X), i)\right] \mathbb{E}_{X \sim p(X|Y=i)}\left[\ell^2(h(X), i)\right]^{\frac{\alpha+1}{\alpha-1}}\right)^{\frac{\alpha-1}{\alpha}}$$

$$\leq \left(\sum_i^k q(i) \frac{q(i)^\alpha}{p(i)^\alpha}\right)^{\frac{1}{\alpha}} \sum_i^k q(i) \mathbb{E}_{X \sim p(X|Y=i)}\left[\ell^2(h(X), i)\right]^{1-\frac{1}{\alpha}} \mathbb{E}_{X \sim p(X|Y=i)}\left[\ell^2(h(X), i)\right]^{1+\frac{1}{\alpha}}$$

$$\tag{14}$$

where the first inequality follows from Hölder's inequality, the second one follows from Jensen's inequality and the fact that the loss is in $[0, 1]$ as well as the fact that the exponentiation function is convex in this region. Moreover, since $1 + \frac{1}{\alpha} \geq 1$ and upper bound for the loss square, $l(\cdot, \cdot)^2 \leq 1$, then;

$$\mathbb{E}_{\mathbb{P}}\left[w(Y)^2 \ell^2(h(X), Y)\right] \leq \left(\sum_i^k q(i) \frac{q(i)^\alpha}{p(i)^\alpha}\right)^{\frac{1}{\alpha}} \sum_i^k q(i) \mathbb{E}_{X \sim \mathbb{P}|Y=i}\left[\ell^2(h(X), i)\right]^{1-\frac{1}{\alpha}}$$

which gives bound on the second moment of weighted loss.

### B.7 SLIGHT DRIFT FROM THE LABEL SHIFT

**Drift in label shift assumption:** If the label shift approximation is slightly violated, we expect the generalizing bound to deviate from the statement in the Theorem. 1. Define $d_e(q||p) := \mathbb{E}_{(X,Y) \sim \mathbb{Q}}\left[\left|1 - \frac{p(X|Y)}{q(X|Y)}\right|\right]$ as the deviation form label shift constraint which is zero in label shift setting.

**Remark 1 (Drift in Label shift assumption)** *If the label shift assumption slightly violates, for the true importance weights* $\omega(x, y) := \frac{q(x,y)}{p(x,y)}$, *the* RLLS, *with high probability generalizes as;*

$$\mathcal{L}(\widehat{h}_{\widehat{w}}, \omega) - \mathcal{L}(h^*; \omega) \leq \epsilon_{\mathcal{G}}(n_p, \delta) + (1-\lambda)\|\theta\|_2 + \lambda \epsilon_\theta(n_p, n_q, \|\theta\|_2, \delta) + 4(1-\lambda) d_e(q||p)$$

Consider the case where the Label shift assumption is slightly violated, i.e., for each covariate and label, we have $p(x|y) \simeq q(x|y)$, resulting importance weight $\omega(x, y) := \frac{q(x,y)}{p(x,y)}$ for each covariate and label. Similar to decomposing in Eq. 9, we have

$$\mathcal{L}(\widehat{h}_{\widehat{w}}; \omega) - \mathcal{L}(h^\star; \omega) = \underbrace{\mathcal{L}(\widehat{h}_{\widehat{w}}; \omega) - \mathcal{L}(\widehat{h}_{\widehat{w}})}_{(c)} + \underbrace{\mathcal{L}(\widehat{h}_{\widehat{w}}) - \mathcal{L}_n(\widehat{h}_{\widehat{w}})}_{(b)} + \underbrace{\mathcal{L}_n(\widehat{h}_{\widehat{w}}) - \mathcal{L}_n(\widehat{h}_{\widehat{w}}; \widehat{w})}_{(a)}$$

$$+ \underbrace{\mathcal{L}_n(\widehat{h}_{\widehat{w}}; \widehat{w}) - \mathcal{L}_n(h^\star; \widehat{w})}_{\leq 0} + \underbrace{\mathcal{L}_n(h^\star; \widehat{w}) - \mathcal{L}_n(h^\star)}_{(a)} + \underbrace{\mathcal{L}_n(h^\star) - \mathcal{L}(h^\star)}_{(b)} + \underbrace{\mathcal{L}(h^\star) - \mathcal{L}(h^\star; \omega)}_{(c)}$$

$$\tag{15}$$

where the desired excess risk is defined with respect to $\omega$. The differences between Eq. 15 and Eq. 9 are in a new term $(c)$ as well as term $(a)$. The term $(b)$ remains untouched.

**Bound on term** $(c)$     For any $h$, the two contributing components in $(c)$, i.e., $\mathcal{L}(h;\omega)$ and $\mathcal{L}(h)$ are as follows;

$$\mathcal{L}(h) = \mathbb{E}_{(X,Y)\sim\mathbb{P}}\left[w(Y)\ell(Y,h(X))\right], \text{ and } \mathcal{L}(h;\omega) = \mathbb{E}_{(X,Y)\sim\mathbb{Q}}\left[\ell(Y,h(X))\right] = \mathbb{E}_{(X,Y)\sim\mathbb{P}}\left[\omega(X,Y)\ell(Y,h(X))\right]$$

For their deviation we have

$$\begin{aligned}
\mathcal{L}(h;\omega) - \mathcal{L}(h) &= \mathbb{E}_{(X,Y)\sim\mathbb{P}}\left[\left(\omega(X,Y) - w(Y)\right)\ell(Y,h(X))\right] \\
&= \mathbb{E}_{(X,Y)\sim\mathbb{P}}\left[\left(\frac{q(X,Y)}{p(X,Y)} - \frac{q(Y)}{p(Y)}\right)\ell(Y,h(X))\right] \\
&= \mathbb{E}_{(X,Y)\sim\mathbb{Q}}\left[\left(1 - \frac{p(X|Y)}{q(X|Y)}\right)\ell(Y,h(X))\right] \le d_e(q||p) := \mathbb{E}_{(X,Y)\sim\mathbb{Q}}\left[\left|1 - \frac{p(X|Y)}{q(X|Y)}\right|\right]
\end{aligned}$$

Where in the last inequality we deploy Cauchy Schwarz inequality as well as loss is in $[0,1]$ and hold for $h \in \mathcal{H}$. It is worth noting that the expectation in $d_e(q||p)$ is with respect to $\mathbb{Q}$ and does not blow up if the supports of $\mathbb{P}$ and $\mathbb{Q}$ do not match

**Bound on term** $(a)$     For any $h \in \mathcal{H}$, similar to the derivation in Eq. 10 we have

$$|\mathcal{L}_n(h) - \mathcal{L}_n(h;\widehat{w})| \le \|\widehat{w} - w\|_2 \le \|\lambda\widehat{\theta} - \theta\|_2 \le (1-\lambda)\|\theta\|_2 + \lambda\|\widehat{\theta} - \theta\|_2$$

The previous weight estimation analysis does not directly hold for this case where the label shift is slightly violated, but with a few modification we provide an upper-bound on the error. Given a classifier $h_0$

$$\begin{aligned}
q_{h_0}(Y = i) &= \sum_j q(h(X) = i|Y = j)q(j) \\
&= \sum_j \left(q(h_0(X) = i|Y = j) - p(h_0(X) = i|Y = j)\right)q(j) + \sum_j p(h_0(X) = i|Y = j)q(j) \\
&= \sum_j \left(q(h_0(X) = i|Y = j) - p(h_0(X) = i|Y = j)\right)q(j) + \sum_j p(h_0(X) = i, Y = j)w_j \\
&= \mathbb{E}_{(X,Y)\sim\mathbb{Q}}\left[p(h_0(X) = i)\left(1 - \frac{p(X|Y)}{q(X|Y)}\right)\right] + \sum_j p(h_0(X) = i, Y = j)w_j
\end{aligned}$$

where $p(h_0(X) = i) = q(h_0(X) = i)$, resulting;

$$q_{h_0} = b_e + Cw$$

where we drop the $h_0$ in both $b_e$ and $C$. Both the confusion matrix $C$ and the label distribution $q_{h_0}$ on the target for the black box hypothesis $h_0$ are unknown and we are instead only given access to finite sample estimates $\widehat{C}_{h_0}, \widehat{q}_{h_0}$. Similar to previous analysis we have

$$b := q - C\mathbf{1} = C\theta$$

with corresponding finite sample quantity $\widehat{b} = \widehat{q} - \widehat{C}\mathbf{1}$. Similarly to the analysis when there was no violation in label shift assumption, we have $\Upsilon(\theta') = \|C\theta' - b - b_e\|_2$ and the solution to Eq. 3 satisfies;

$$\Upsilon(\widehat{\theta}) \le \inf_{\theta'\in\mathcal{R}^k}\{\Upsilon(\theta') + 2\Delta_C\|\theta'\|_\alpha\} + 2\Delta_b + 2\|b_e\|_2. \tag{16}$$

We can simplify the upper bound by setting $\theta' = \theta$. We then have

$$\Upsilon(\widehat{\theta}) = \|C\widehat{\theta} - b - b_e\|_2 = \|C\left(\widehat{\theta} - \theta\right)\|_2 \le 2\Delta_C\|\theta\|_2 + 2\Delta_b + 2\|b_e\|_2 \le 2\Delta_C\|\theta\|_2 + 2\Delta_b + 2d_e(q||p)$$

resulting in

$$\|\widehat{\theta} - \theta\|_2 \le \frac{1}{\sigma_{min}}\left(2\Delta_C\|\theta\|_2 + 2\Delta_b + 2d_e(q||p)\right)$$

