# OpenReview forum: "Regularized Learning for  Domain Adaptation under Label Shifts"
_ICLR.cc/2019/Conference_

### Official Review · AnonReviewer1 · 2018-11-01
**A contribution for a rather unstudied problem with new theoretical results used in the implementation, improved empirical results - however state-of-the-art section is incomplete leading to a lack of baselines, lack of comparison with the close related work of Lipton et al.'18**

**Rating:** 6
**Confidence:** 4

**Review:**

This paper presents a new contribution for a largely understudied problem of label shift (also called target shift), a situation occurring when the class proportions vary between the training and test sets. The proposed contribution builds upon a recent work on the subject by Lipton et al., 2018 and addresses several of its weaknesses. The paper also gives several improved generalisation bounds w.r.t. that of Lipton et al. that are further used as guidelines to tune the regularisation parameter based on the size of source and target samples. Finally, the empirical results show that the proposed algorithm outperforms that of Lipton et al. especially in cases where the shift in proportions becomes quite important.

*Pros:
   - A work in an area with very view contributions and a certain lack of theoretical results
    -Theoretical results that are actually used in the algorithmic implementation and that allow to define the regularisation parameter based on the size of the available samples
    -Improved empirical results


*Cons:
    -An incomplete state-of-the-art section that does not cite several important contributions on the subject;
    -Lack of baselines due to the incomplete state-of-the-art section;
    -Lack of clear comparison with Lipton et al. both in terms of the proposed method and the obtained theoretical guarantees.


*Detailed comments:
This paper is rather interesting and well-written.

I have several major concerns regarding this paper. They can be summarised as follows:

    There is an important part of literature review on target shift that is missing in this paper. Even though, the paper mentioned the work of Chang, 2005 and Zhang, 2013, it completely ignores several other highly relevant methods such as [1,2]. These works also propose algorithms that allow to estimate class proportions that vary between training and test data. This estimation can then be used for cost-sensitive learning to correct the target shift. The paper should mention this work and add the corresponding methods to the baselines for comparison.

    Several statements that justify the contribution of this paper are unsupported. For instance, the paper states that the estimator obtained with the inverse of the confusion matrix can be arbitrary bad when the sample size and/or the singular values are small. However, this exact dependence can be found in Lemma 1 for the proposed contribution also! This is repeated in the beginning of Section 2.2 to justify the regularised version of the estimator but once again no evidence was provided to support the claim. The obtained bound for the regularised algorithm also has these two terms and thus it is not clear why the regularised algorithm is supposed to work better.

    The paper may want to clearly state the differences between the proposed algorithm and that of Lipton et al. and also between the obtained error bounds. The paper states that it achieves a k*log(k) improvement over Lipton et al. bounds but as fair as I can see this improvement is achieved only when h_0 is an ideal estimator. Furthermore, Lipton et al.’s bounds are linear in k while the proposed bounds replace this term with log(k/delta) so that when \delta is small, ie the bound holds with high probability, the bound becomes much worse. I would suggest to add a brief discussion on the relationship between the two to better highlight the original contribution of the paper.

    The proofs are quite badly written with many lacking results used to move from one inequality to another. For instance, Lemma 2 is proved using the theorem 1.4[Matrix Bernstein] and dilation technique from Tropp but it is not clear which results the authors are using in particular; Theorem 1.4 is related to the largest eigenvalue of the sum of matrices while the authors obtain an inequality for the norm of the sum without any further comment on how this transition was made. Also, I do not see why delta is smaller than 1/2 in Lemma 2.


*Minor comments:

   - p.1: expected have -> expected to have
   - p.4: we are instead only gave access -> given access to
   - I do not understand Figure 1. Should it be n_q*n_p on the y axis ?
   - The inequality for n_q next to Figure 1 is derived from the bound (6). Why it is independent of k?
   - Why the authors choose to the black box predictor h0 to be a two-layer fully connected neural? Is there any particular reason to use this classification model?

[1] Class Proportion Estimation with Application to Multiclass Anomaly Rejection, AISTATS14
[2] Mixture Proportion Estimation via Kernel Embeddings of Distributions, ICML16

---

> ### Author Response · Authors · 2018-11-16
> **A contribution for a rather unstudied problem with new theoretical results used in the implementation, improved empirical results**
>
> We thank the reviewer for the detailed review and pointers to related works. Please see our detailed answers to your comments below.
>
> 1) Related literature:
>
> We appreciate the reviewer's pointers to related works based on an anomaly detection framework. The two mentioned papers as well as Blanchard et al. 2010, are considered as pioneer works in this area and do make a strong theoretical contribution. We added them to our related works along with a discussion. These methods employ a function class to estimate the class proportions where they require the knowledge of the VC dimension of this function class for weight estimation task. It has elegant theoretical results under mutual irreducibility assumption and derive asymptotic guarantees on the weight estimation which depends on the VC dimension of the mentioned class.  However, as we understand these approaches when a class of deep neural networks with unknown (or at least vacuous) VC dimension is deployed, the comparison is empirically prohibited. Moreover, these methods require solving additional computationally expensive optimization problems in their inner loop weight estimation which are intractable for large function classes such as deep networks. We again appreciate the reviewer's suggestion and added a discussion about these methods in our paper.
>
> 2) How the estimator obtained with the inverse of the estimated confusion matrix can be arbitrarily bad and ours is not:
>
> Technically, estimating \hat{w} using the inverse of the estimated confusion matrix \hat{C} can be arbitrarily bad since the \hat{C} can be arbitrary close to a singular matrix.  In the low sample setting when the smallest singular value \sigma_\min of the true confusion matrix C is small, this issue gets amplified (reflected in the bound). In fact, Lipton et al. 2018 even require that the number of samples used to estimate the confusion matrix (the number samples in the source domain) has to be larger than O(1/\sigma_\min^2). That means, they do NOT offer guarantees for the small sample regime.
>
> Other than being a very large lower bound, this criterion is unrealistic to check since \sigma_\min is not known a priori. They require this constraint to make sure that with high probability the estimated confusion matrix \hat{C} is bounded away from being singular. In contrast, we deploy the principles of statistical linear inverse problems and propose a weight estimation approach which does not require inverting \hat{C} and estimate the importance-weights \hat{w} by solving a convex optimization problem.
>
> We have clarified our improvements compared to BBSL including minimum sample complexity and k \log k  in the introduction and paragraphs after Lemma 1. We incorporated this comment in our main draft and stated that the \hat{C} can be arbitrarily close to a singular matrix.
>
> 3) Comparison with Lipton et al ( factor k improvement ). and improvement only achieved when h_0 is ideal:
> We added a more detailed and direct comparison between our weight estimator and Theorem 3 in Lipton et al.  in the discussion after Lemma 1.
>
> First regarding the factor k improvement: With respect to the dependence on \delta, a slight confusion might have resulted from the way Theorem 3 in Lipton et al. was stated compared to ours. The authors did not write the statement in the form “with probability at least 1- \delta” but rather with terms dependent on k, n_q, and n_p. Translating the theorem to have a 1-\delta type guarantee results in a dependence of delta that is exactly like ours.
>
> Regarding ideal h_0:  h_0 need not be an ideal estimator, neither in Lipton et al.’s nor in our paper. The dependence on h_0 is implicit via \sigma_{min}. We have emphasized this fact a bit more in the paper in the second paragraph after Lemma 1.
>
> 4) Lemma 2 and its connection to Theorem 1.4 in Tropp 2012, as well as \delta<0.5 assumption in Lemma 3:
> We added the definition of the norm by which we meant the spectral norm, equivalent to the largest eigenvalue of the matrix. Moreover, we also added more detailed links to theorems and results in Tropp 2012.
>
> Regarding delta<1/2 in Lemma 3: This was originally just a technical assumption (and natural, since usually, you want the bound to hold with probability above 0.5) to simplify the bound and make the dependence on n more transparent. In order to avoid confusion, we now removed it from our lemma statement. The change ultimately transfers to some change of universal constants in the upper bounds which we neglect by using the O-notation.
>
> 5) Thanks for the detailed comments. We have corrected the typos
> Re: Figure 1: The labeling is actually correct as is. The dependence on k of the bound in Theorem 1 is in the log factors which were neglected in the lower bound to simplify the presentation
> Re two layer fully connected neural: We made it more clear in the revised version that the bounds are independent of h_0 and that the dependence of the bound on h_0 is only via \sigma_min.

---

> > ### Comment · AnonReviewer1 · 2018-11-28
> > **About baselines**
> >
> > Thank you for your answers.
> >
> > The code for [1] can be actually found here http://web.eecs.umich.edu/~cscott/code.html. There are at least two methods that authors can use as baselines that can be deployed on real data.

---

> > > ### Author Response · Authors · 2018-12-03
> > > **memory issue**
> > >
> > > We would like to thank you for providing the link to the code. Inspired by your suggestion to analyze [1], we deployed the code provided by the authors and ran it for data with dimensionality equal to 700 (similar to MNIST).
> > > In our experiments, we found that the largest number of samples for which we could feasibly run their code without memory halt was 11k.  A similar observation was also reported in Lipton et al. 2018 where the authors were not able to apply the kernel mean matching approach of Zhang et al., 2013 on datasets of size larger than 8k samples (we are not aware of the type of machine they used in order to report this number). This hinders the application of these methods to real-life machine learning problems where the data size will be often above 100K. The aim of our paper is explicitly to find a method that works for general large-scale (training) datasets, and thus due to the memory issue, we do not plan to provide a comparison to these baselines in all of our plots. However, we can conduct a comparative study on a smaller dataset in a separate experiment and include it in the paper. We again appreciate your helpful comment, and we are working on adding a detailed explanation of this study to our paper.

---

### Official Review · AnonReviewer2 · 2018-11-01
**Improved estimators for correcting label shifts, but experiments can be improved**

**Rating:** 6
**Confidence:** 4

**Review:**

- The authors consider the problem of learning under label shifts, where the label proportions p(y) and q(y) of the training and test distributions differ, while the conditionals p(x|y) and q(x|y) are equal. They build upon the work by Lipton et al. 18 on estimating label proportion weights q(y)/p(y) using the confusion matrix, by proposing an improved estimator with regularization. They show that their estimator provides better weight estimates compared to the unregularized version, and it also gives better prediction accuracies under large label shift scenarios.

- One question I have about this approach is the choice of h in the confusion matrix estimation. Since the theory holds for any fixed hypothesis h, is there any guidance on how we should pick h? The authors seem to use the same model class for the weight estimation and predictions in the experiments. How would using a simpler h for weight estimation (e.g., linear logistic regression) affect the results presented here?

- The Dirichlet shifts described with only the parameter alpha is not particularly intuitive in conveying the size of shifts. The CIFAR10 and MNIST datasets contain about 6000 examples per class. How would a large shift with alpha=0.01 change the distribution, especially for the smallest class how many samples are retained? This can help the readers judge when the correction of label shifts are helpful.

- To clarify, in the experiments for Figure 4 using Minority-Class shifts, with p=0.001, is it true that there are less than 100 training examples for each of the minority classes in the training set? This seems like a very extreme shift.

- I also have trouble understanding Figure 3. RESNET-18 should give >90% accuracy on the original CIFAR10, but in 3b we see accuracies around 75% for small shifts. Also how is the F1-score in 3c computed? Is it micro-averaged or macro-averaged F1? Either way an F1 score below 20% is very low for the unweighted classifier, since RESNET-18 should give fairly good classification accuracy on each class separately if it has >90% overall accuracy.

- The paper is quite solid in motivating the need for better weight estimators for reweighing label proportions and their derivations, and manage to show improvements over the unregularized estimator. Details on the experiments should be improved to give the readers better ideas on when correcting for label shifts help. Right now it looks like it only helps for cases with fairly extreme shifts.

---

> ### Author Response · Authors · 2018-11-16
> **Improved estimators for correcting label shifts, but experiments can be improved**
>
> We want to thank you for your thoughtful response and detailed questions.
>
> 1) model for black box predictor h_0:
> The minimum singular value of the true C depends heavily on how well h can predict Y (which depends on the model you choose for the respective dataset in question). For example, imagine in binary classification that h predicts uniformly at random. Then all entries of C will be 1/2, C will have rank 1 and thus minimum singular value 0. This not only makes C hard to estimate. The final upper bound on the excess risk (in Theorem 1 and variants) also explicitly depends inversely on the minimum eigenvalue of C. Therefore, the better the black box predictor h_0 is in prediction, the better and more stable the estimation of C and the smaller the upper bound. We hope that the reformulation of the discussion in the second paragraph following Lemma 1 clarifies this point. You may also have a look at Figure 4 which illustrates the influence of the black box prediction accuracy directly.
>
> 2) alpha for Dirichlet shift, number of samples for alpha = 0.01:
> In the Dirichlet shift, the probability vector p is sampled (from the simplex) according to a Dirichlet distribution with constant concentration parameters \alpha_1 … \alpha_k = \alpha. The larger \alpha, the more mass is concentrated in the middle of the simplex (more uniform) and the smaller \alpha, the more mass on the edges (more skewed), smaller p for the smallest class. I.e. the smaller \alpha, the bigger the shift. We chose to run experiments on a variety of shifts to see how our method behaves under different shifts, and Dirichlet in particular since it was used in Lipton et al. While the total number of data points is set to 10000, for rather large shifts, it is possible that the smallest class has zero samples and this indeed happens for \alpha = 0.01
>
> 3) Minority-Class shifts with extreme shifts p=0.001:
> Yes, it is an extreme shift case. In our revision, we now use p=0.005 (now Figure 2) to better show when our method would help and outperform both unweighted classifier and Lipton et al. We have two more figures on p=0.01 and p=0.001 to the appendix. We will put more sets of experiments using different sample sizes and values of p in the appendix.
>
> 4) Figure 3 low accuracy for small shifts, micro- vs. macro F-1 score, low total F-1 score:
> Figure 3 shows the case when the target data is Dirichlet shifted with small alpha indicating a larger shift. In this paper, in order to create shifted data with certain label proportions, we do not use full the training set or testing set. In fact, both source and target sets consist of 10000 examples for all \alpha. This may compromise performance but makes the comparison (between shifts) fair since the amount of data is fixed.  We had added more explanation in the revision.
>
> Thanks for the clarifying question, we added a more detailed description of our computation of the F-1 score in the paper now (second paragraph of 3.2.). The F-1 score is macro averaged so that when the test data is dominated by only a few classes, F-1 could be very low even though the overall accuracy is high. Note that the standard python package that we use, handles ill-defined cases as follows: When a class is not present but predicted in the target set, the F-1 value of that class is 0 while if there are no predicted examples in an absent class, the F-1 is counted as 1. Our ResNet is trained with fewer samples and has ~75% accuracy, which helps to explain the very low F-score for the extreme shift  \alpha =0.01: in fact there is only one class present in the target set, and any sample that is predicted to belong to one of the non-existent class, results to a zero F-1 value, thus decreasing the macro-averaged F-1 score drastically.
>
> 5) The method only helps with fairly extreme shifts:
> The reason why we compare performances for relatively large shift cases is to demonstrate the advantages of our proposed method over BBSE in Lipton et al., achieving smaller weight estimation errors due to the regularization procedure. As shown in Figure 2(a) and Figure 3(a), as the shifts get larger, the weight estimation error of RLLS increases much more compared to BBSL. The harder the problem (i.e. more extreme shift or/and smaller target sample sizes, source shift rather than target shift), the bigger the advantage of our method compared to the baseline. The goal of our paper is exactly to target the hard regime. However, correcting label shifts should yield better accuracy (compared to the unweighted classifier) in more general cases using either method (RLLS or BBSL). We elaborate on this more in our revision and will also add more experiments on different shift types and CIFAR10 in the final version to give a more complete picture of the regimes where it is helpful.

---

### Official Review · AnonReviewer3 · 2018-11-02
**Interesting Algorithm with Solid Theories**

**Rating:** 7
**Confidence:** 3

**Review:**

Authors proposes a new algorithm for improving the stability of class importance weighting estimation procedure (Lipton et al., 2018) with a two-step procedure. The reparamaterization of w using the weight shift theta and lambda allows authors develop a generalization upperbound with terms rely on theta, sigma and lambda.

The problem of label shift is a known important issue in transfer learning but has been understudied.

The paper is very well written and the algorithm is well-motivated (introducing regularization to avoid the singularity) and post processing step looks sound (using lambda to de-biase). I only have a few minor questions:

1. How realistic it is to assume we have prior knowledge on theta and sigma_min?

2. If I understand correctly, the only experiment where lambda is varied is Sec 3.3? It would be interesting if authors also included BBSE in Sec 3.3 as a baseline.

3. The authors mentioned in the discussion that the generalization guarantee is obtained with no prior knowledge q/p is needed. However, doesn't theta implicitly represent the knowledge in p/q?

------------------------------------------------

I have read authors' comments.

---

> ### Author Response · Authors · 2018-11-16
> **Interesting Algorithm with Solid Theories**
>
> Thank you for your positive comments about the paper. Please see detailed answers to your questions below
>
> 1. How realistic it is to assume we have prior knowledge on theta and sigma_min?
>
> 1) Prior knowledge on upper bound of theta and lower bound of sigma_min. Our bounds hold for all choices of \lambda, \theta and \sigma_\min. In general, \theta is unknown. However, it is reasonable to assume that we only want to be robust against shifts up to a certain \theta_\max, so that we essentially consider only the set of \theta with norm up to \theta_\max.
> As to \sigma_\min in practice, although we may not know the \sigma_\min of the true confusion matrix, we can estimate it using the empirical confusion matrix. We have clarified that in algorithm 1 and added a clarifying discussion about both these matters in the paragraphs following Theorem 1.
>
> 2. If I understand correctly, the only experiment where lambda is varied is Sec 3.3? It would be interesting if authors also included BBSE in Sec 3.3 as a baseline.
>
> 2) We appreciate the reviewer's comment on the BBSE experiment. Upon the reviewer’s suggestion, we also ran BBSE for the experiment in Sec 3.3 and added the corresponding curves to Figure 5.
>
> 3. The authors mentioned in the discussion that the generalization guarantee is obtained with no prior knowledge q/p is needed. However, doesn't theta implicitly represent the knowledge in p/q?
>
> 3) We apologize for the lack of clarity in the mentioned statement. As the reviewer correctly observed, our generalization bound depends on theta. We clarified this statement to emphasize that, in contrast to prior methods, e.g., Chan & Ng, 2005 and Storkey, 2009, our importance weighting algorithm does not require any prior knowledge of theta and results in a generalization bound which depends on the theta.

---

### Author Response · Authors · 2018-11-16
**General reply to the reviewers and  the area chair**

We would like to thank the reviewers and area chair for their thoughtful responses to our paper. We are grateful to each of you for the suggestions that helped us to improve the clarity of the presentation. To improve the flow and clarity of the presentation, we have restructured paper in various places. We have moved some of the experiments to the appendix and added new ones (some of them to the main text) that help to clarify the reviewers' questions. We added and reformulated clarifying discussions about our results and how they compare with baselines.

We have run more experiments which aim to clarify the regime in which our procedure has advantages compared to other label shift correcting methods. We aim to add more figures in the appendix in a potential camera ready version for different shifts parameters on CIFAR10 to present an even more complete picture.

Please find individual replies to each of the reviews in the respective threads.

---

### Meta-Review · Area_Chair1 · 2018-12-17

**Confidence:** 5
**Recommendation:** Accept (Poster)

**Metareview:**

The paper gives a novel algorithm for transfer learning with label distribution shift with provably guarantees. As the reviewers pointed out, the pros include: 1) a solid and motivated algorithm for a understudied problem 2) the algorithm is implemented empirically and gives good performance. The drawback includes incomplete/unclear comparison with previous work. The authors claimed that the code of the previous work cannot be completed within a reasonable amount of time. The AC decided that the paper could be accepted without such a comparison, but the authors are strongly urged to clarify this point or include the comparison for a smaller dataset in the final revision if possible.